# A novel GSK3-regulated APC:Axin interaction regulates Wnt signaling by driving a catalytic cycle of efficient βcatenin destruction

**Mira I Pronobis[1], Nasser M Rusan[2], Mark Peifer[1,3,4]***

[1]Curriculum in Genetics and Molecular Biology, University of North Carolina at Chapel Hill, Chapel Hill, United States; [2]Cell Biology and Physiology Center, National Heart, Lung, and Blood Institute, Bethesda, United States; [3]Biology Department, University of North Carolina at Chapel Hill, Chapel Hill, United States; [4]Lineberger Comprehensive Cancer Center, University of North Carolina at Chapel Hill, Chapel Hill, United States

**Abstract** APC, a key negative regulator of Wnt signaling in development and oncogenesis, acts in the destruction complex with the scaffold Axin and the kinases GSK3 and CK1 to target βcatenin for destruction. Despite 20 years of research, APC's mechanistic function remains mysterious. We used FRAP, super-resolution microscopy, functional tests in mammalian cells and flies, and other approaches to define APC's mechanistic role in the active destruction complex when Wnt signaling is off. Our data suggest APC plays two roles: (1) APC promotes efficient Axin multimerization through one known and one novel APC:Axin interaction site, and (2) GSK3 acts through APC motifs R2 and B to regulate APC:Axin interactions, promoting high-throughput of βcatenin to destruction. We propose a new dynamic model of how the destruction complex regulates Wnt signaling and how this goes wrong in cancer, providing insights into how this multiprotein signaling complex is assembled and functions via multivalent interactions.

*For correspondence: peifer@unc.edu

**Competing interests:** The authors declare that no competing interests exist.

**Reviewing editor**: Helen McNeill, The Samuel Lunenfeld Research Institute, Canada

## Introduction

The Wnt signaling pathway is one of the most critical in animals, controlling both cell proliferation and fate (*Clevers and Nusse, 2012*). Deregulated Wnt signaling plays roles in several cancers (*Polakis, 2007*; *Kandoth et al., 2013*); most strikingly, mutations in Adenomatous polyposis coli (APC), a key negative regulator of Wnt signaling, initiate ~80% of colon cancers. Although APC was identified in 1991, its mechanistic role in Wnt regulation remains mysterious.

Wnt signaling regulates levels of the transcriptional co-activator βcatenin (βcat; *Clevers and Nusse, 2012*). Like other powerful signaling pathways driving development and oncogenesis, animals evolved dedicated machinery to keep the Wnt pathway off in the absence of signal. In the OFF-state, βcat levels are kept low by the action of the multiprotein destruction complex, which includes APC (*Figure 1A*), the scaffold Axin (*Figure 1A*) and the kinases GSK3 and CK1. In the current model, βcat is recruited into the destruction complex by binding APC or Axin and sequentially phosphorylated by CK1 and GSK3. This creates a binding site for the E3-Ligase SCF$^{\beta TrCP}$ and βcat is ubiquitinated and destroyed. Wnt ligand binding to the Wnt receptor inhibits the destruction complex, through a series of events whose order and relative importance remains unclear (*MacDonald and He, 2012*). βcat levels rise and it activates Wnt target genes.

**eLife digest** An embryo starts off as a small ball of stem cells, each of which has the potential to become any type of cell in the body. Adult organs and tissues also contain small numbers of stem cells that can replace old or damaged cells. In both of these processes, stem cells need to 'decide' when they should start to change into a more specialized cell type, and which cell fate to choose (e.g., liver cell vs kidney cell). A signaling pathway involving Wnt proteins helps to direct many of these decisions. But if the 'Wnt signaling pathway' becomes activated at the wrong time, it can lead to cancer. For example, the first step in development of colon cancer is the inappropriate activation of Wnt signaling, and is most often caused by mutations in the gene that encodes a protein called APC. The APC protein is a tumor suppressor and normally inhibits Wnt signaling. However, even after over 20 years of effort, it remains largely mysterious how APC does this.

APC is known to work with another protein called Axin as part of a large protein machine. This protein complex performs one of the first steps in a process that ultimately marks a key component of the Wnt signaling pathway for destruction. Pronobis et al. have now used a range of techniques to define APC's role in this so-called 'destruction complex'. This analysis revealed the internal structure of a complex made from APC and Axin, and showed that cable- and sheet-like assemblies of Axin were intertwined with APC cables. Further experiments then revealed how APC and Axin proteins are added into or leave these complexes, and showed that this is critical for this protein machine to work.

Pronobis et al.'s data also suggest that APC plays two roles, which make the destruction complex more efficient. Firstly, it can interact with Axin via two separate interaction sites that help to assemble the destruction complex. Secondly, specific features in APC allow it to interact with a third protein (called GSK3), which can then regulate how APC interacts with Axin. One of the next challenges will be to uncover how APC helps to transfer the components of Wnt signaling to the next step of their destruction, and to clear up the role played by GSK3.

Despite this attractive model, several key issues remain. APC and Axin are destruction complex core components. Both are protein interaction hubs, combining folded domains with intrinsically disordered regions carrying peptide motifs that bind protein partners. APC was initially thought to be the scaffold templating βcat phosphorylation but subsequent work revealed that Axin plays this role (*Hart et al., 1998*; *Ha et al., 2004*). Axin has binding sites for βcat, APC, GSK3, CK1 and PP2A, and self-polymerizes via its DIX domain (*Figure 1A*), which allows it to locally increase the concentration of proteins mediating βcat phosphorylation (*Ikeda et al., 1998*; *Fagotto et al., 1999*; *Liu et al., 2002*; *Fiedler et al., 2011*). Thus while APC is essential for destruction complex function, its role in the complex remained a mystery.

APC combines an N-terminal Armadillo repeat domain (Arm rpts) with a more C-terminal intrinsically disordered region; each contains binding sites for multiple partners (*Figure 1A*). The Arm rpts bind cytoskeletal regulators (*Nelson and Nathke, 2013*), and also help regulate Wnt signaling but the relevant binding partner(s) for this is unclear (*Roberts et al., 2012*). APC's SAMP repeats bind Axin's RGS region, helping mediate destruction complex assembly. Both the Arm rpts and SAMPs are essential for APC function. APC also has multiple βcat binding sites, the 15 and 20 amino acid repeats (15Rs and 20Rs), which have different affinities for βcat, and act redundantly to sequester βcat and fine tune Wnt signaling (*Liu et al., 2006*; *Roberts et al., 2011*). However, surprisingly, these binding sites are not essential for APC's mechanistic role in βcat destruction (*Yamulla et al., 2014*).

Recent work led to significant insights into how Wnt signaling turns off the destruction complex. Axin turnover and Dvl-Axin hetero-polymerization may inhibit destruction complex function by disassembling the destruction complex (*Schwarz-Romond et al., 2007*; *Fiedler et al., 2011*). However, recent studies revealed that the destruction complex remains intact for some time after Wnt signals are received, and can continue to phosphorylate βcat (*Hernandez et al., 2012*; *Li et al., 2012b*; *Kim et al., 2013*). Thus these studies suggest that Wnt signaling leads to either reduction in the rate of βcat phosphorylation by the destruction complex, or that it inhibits a key regulated step after phosphorylation, perhaps transfer to the E3 ligase, thus turning down an intact destruction complex, leading to high levels of βcat.

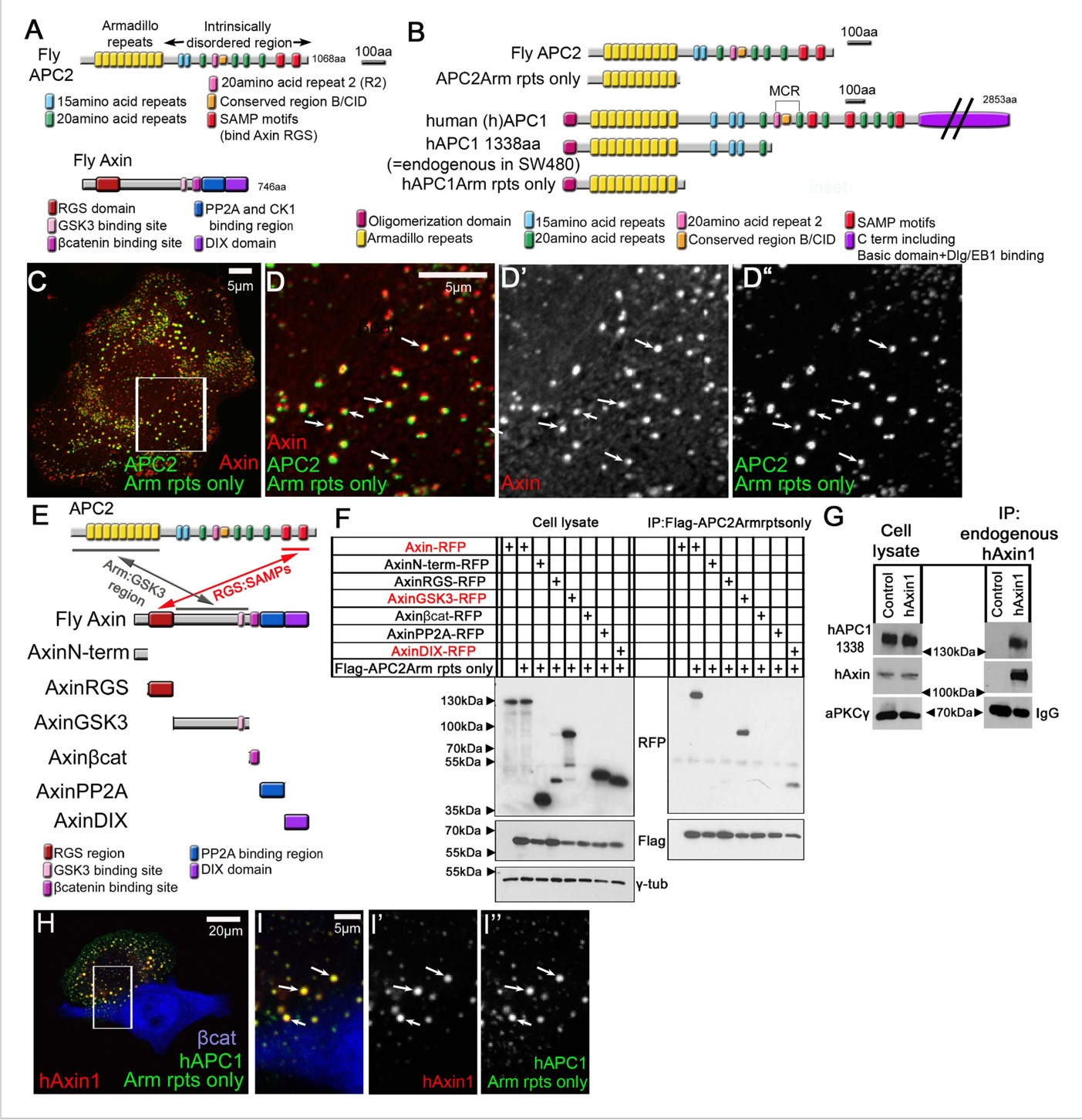

**Figure 1**. APC2's Arm rpts provide a second means of interacting with the Axin complex. (**A**) Fly APC2 and Axin. (**B**) Constructs used. hAPC1-1338 = the endogenous truncated hAPC1 in SW480 cells. (**C** and **D**) SW480 cells coexpressing GFP-APC2Arm rpts only and Axin-RFP, which localize adjacent to one another (arrows). (**D**) Insets = box in (**C**). (**E**) Known and novel APC:Axin interaction sites (top) and Axin constructs (bottom). (**F** and **G**) IPs from SW480 cells. (**F**) APC's Arm rpts coIP with Axin's middle region that contains the GSK3 binding site. Axin's DIX domain was weakly detected. The βcat binding site fragment was not detected in either immunoblots or immunofluorescence (not shown), suggesting rapid degradation. (**G**) IP of endogenous hAxin1 or control Ig. Truncated endogenous hAPC1-1338 coIPs with hAxin1 at endogenous levels. (**H**) GFP-hAPC1Arm rpts only and hAxin1-RFP colocalize in puncta (arrows). (**I**) Insets = box in (**H**).

*Figure 1. continued on next page*

*Figure 1. Continued*

The following figure supplements are available for figure 1:

**Figure supplement 1**. Assessing levels of over-expression of Axin and APC.

**Figure supplement 2**. Human APC's Arm repeat domain colocalizes with Axin.

In contrast, less attention has been devoted to the active destruction complex when Wnt signaling is off, the state most relevant to what happens in colon tumors. Strikingly, these tumors lack wild-type APC but instead invariably express a truncated APC protein retaining some but not all functional domains—the reason for this remains controversial (*Albuquerque et al., 2002*). Further, while tumor cells do not effectively target βcat for destruction, it is less clear at what step destruction is blocked by APC truncation. In the simplest model, mutant destruction complexes wouldn't template βcat phosphorylation, but Axin's ability to do this in vitro in APC's absence cast doubt on this (*Ha et al., 2004*). Further, cancer cells with truncated APC have high levels of phosphorylated βcat, suggesting that the step blocked is after βcat phosphorylation (*Yang et al., 2006*; *Kohler et al., 2008*). One intriguing study proposed that APC acts in the destruction complex to protect βcat from dephosphorylation and then targets it to the E3 ligase (*Su et al., 2008*). However phospho-βcat is elevated in APC-mutant cell lines, despite loss of APC function suggesting that protecting βcat from dephosphorylation is not APC's only role in the destruction complex (*Yang et al., 2006*). Thus APC's key mechanistic function in the destruction complex remains largely unknown.

In exploring APC's mechanism of action when Wnt signaling is inactive, we recently focused on two conserved binding sites in APC, 20R2 (R2) and motif B (B; this is also known as the catenin interaction domain = CID; *Kohler et al., 2009*; *Roberts et al., 2011*; *Schneikert et al., 2014*; *Choi et al., 2013*; *Figure 1A*). R2 is related to the 20R βcat binding sites, but lacks a key interacting residue and cannot bind βcat (*Liu et al., 2006*). B is immediately adjacent to R2—its function was initially unknown, though it was recently shown to bind α-catenin, and thus play a role in Wnt regulation (*Choi et al., 2013*). Strikingly, although other 20Rs are individually dispensable, both R2 and B are essential for the destruction complex to target βcat for destruction (*Kohler et al., 2009*; *Roberts et al., 2011*; *Schneikert et al., 2014*). Our data further suggested that R2/B negatively regulates APC/Axin interaction, a somewhat surprising role for an essential part of the destruction complex.

This prompted us to broaden our analysis. Most destruction complex models, even those that consider the kinetics of initial destruction complex assembly (*Lee et al., 2003*), portray it as a static entity that binds, phosphorylates and hands off βcat. Recent work prompted us to consider an alternative hypothesis, viewing the destruction complex as a complex multiprotein entity whose assembly, structure and dynamics are key to its function in maintaining low βcat levels when Wnt signaling is off. To test this hypothesis, we used FRAP, super-resolution microscopy and other approaches to explore the structure and dynamics of the destruction complex. This provided new insights into APC's mechanism of action, providing evidence that it plays two roles inside the active destruction complex: (1) APC promotes efficiency of the destruction complex by enhancing complex assembly through two separate APC:Axin interaction sites, one of which is novel, (2) the novel APC: Axin interaction is dynamic, and regulation of this interaction by GSK3 acting via motifs R2 and B is essential to send phosphorylated βcat to destruction. More broadly, our data also provide insights into how intrinsically disordered regions assist in the assembly and dynamics of multiprotein signaling complexes.

## Results

### Goal and system used

Our goal is to define APC's mechanistic role in βcat regulation when Wnt signaling is off. In recent work, we discovered that two motifs in APC's intrinsically unstructured region, R2 and B, are essential for promoting βcat destruction in human cells and in *Drosophila* (*Roberts et al., 2011*). Here we sought to define the mechanism by which these motifs and APC itself act, using as a model SW480

colon cancer cells. These cells have high βcat levels, as they lack wildtype human APC1 (hAPC1) and instead express a truncated APC1 protein ending before the mutation cluster region (MCR; *Figure 1B*). SW480 cells also express human APC2 (*Maher et al., 2009*), but this is not sufficient to help mediate βcat destruction. We express in these cells fly APC2, the homolog of hAPC1, a full length APC that shares all conserved regions important for Wnt regulation with hAPC1 but is significantly smaller in size (*Figure 1B*). Fly APC2 effectively reduces βcat levels in SW480 cells (*Roberts et al., 2011*), and thus can interact with all human destruction complex proteins needed to target βcat for degradation.

There is abundant evidence that the functional destruction complex is a multimer of the individual destruction complex proteins. One important underpinning of this idea is that Axin oligimerizes via self-polymerization of the DIX domain, and this multimerization is critical for its Wnt regulatory function (*Kishida et al., 1999*; *Schwarz-Romond et al., 2007*). Endogenous Axin forms small puncta in cultured cells and when overexpressed these puncta become more prominent, in a DIX-domain dependent fashion (*Fagotto et al., 1999*; *Faux et al., 2008*; *Figure 1—figure supplement 1A*). The level of Axin over-expression needed to trigger more prominent Axin puncta is not dramatic—for example, treatment of SW480 cells with tankyrase inhibitors increased levels of AXIN1 3-5x and AXIN2 5-20x and this was sufficient to trigger formation of Axin puncta (*de la Roche et al., 2014*). Axin puncta are dynamic multiprotein complexes that can recruit APC and other destruction complex proteins, and previous data from many labs are consistent with the idea that the puncta can serve as useful models of the smaller endogenous destruction complexes, based on correlations between puncta formation, dynamics, and function in βcat destruction (e.g. *Faux et al., 2008*; *Fiedler et al., 2011*).

We and others previously identified the key structural domains of APC and Axin that are essential for destruction complex function and βcat destruction (e.g. *Roberts et al., 2011*). Our current goal was to define how these proteins' domains function together to facilitate APC and the destruction complex's mechanisms of action. To do so, we used the APC:Axin puncta formed in SW480 cells as a visible and thus measurable read-out to study mechanisms underlying destruction complex structure, assembly, dynamics and function.

Our experiments and those of many earlier investigators used transfected human or in our case fly proteins to study Wnt signaling in cultured mammalian cells (e.g., *Bilic et al., 2007*; *Fiedler et al., 2011*; *Kim et al., 2013*). This strategy likely leads to both variable expression levels between cells and elevated expression relative to endogenous protein. We first investigated variation from cell to cell within a transfection, by quantitating whole cell fluorescence of GFP- or RFP-tagged APC2 or Axin and investigating whether different levels of Axin or APC2 expression altered the ability to down-regulate βcat levels (*Figure 1—figure supplement 1B*). There was a substantial range of APC2 or Axin expression levels among cells (5- to 10-fold). Importantly, βcat levels were substantially reduced at all levels of APC2 or Axin expression, even the lowest levels assessed—this was true for Axin alone, APC2 alone, or Axin plus APC2 (*Figure 1—figure supplement 1B*). In all cases, ability to reduce βcat levels was somewhat diminished at the highest levels of expression (*Figure 1—figure supplement 1B*)—this may be because at very high expression levels, the transfected protein forms non-functional complexes with only a subset of the destruction complex proteins, as was previously suggested (*Lee et al., 2003*).

We next used immunoblotting to get order of magnitude estimates for the level of expression of our transfected constructs relative to the endogenous proteins. We describe the procedure used in detail in the Materials and methods. We began with Axin, to determine the level of over-expression of fly Axin vs endogenous human Axin. Since fly Axin is not recognized by hAxin1 antibodies we did this in two steps, first comparing levels of tagged human Axin1 vs *Drosophila* Axin using antibodies against the epitope tag, and then comparing the levels of the transfected human Axin1 vs the endogenous Axin1 protein in SW480, using antibodies against human Axin1 (*Figure 1—figure supplement 1C*). We then used these ratios and the transfection efficiencies to calculate the average ratio of *Drosophila* Axin:endogenous hAxin1. Transfected Axin accumulated at roughly 80- to 120-fold that of endogenous hAxin1 (*Table 1*). For APC, estimating 'overexpression' was more problematic, as SW480 cells do not accumulate wild-type APC1—instead they accumulate a truncated APC1 ending at amino acid 1338 (*Figure 1B*). We thus used a similar two-step process, comparing levels of tagged human APC1 cloned so as to mimic the truncated APC1 seen in SW480 cells vs tagged *Drosophila* APC2 using antibodies to the epitope, and then levels of tagged truncated human APC1 vs that of the endogenous truncated APC1 protein (*Figure 1—figure supplement 1D*). This ratio was roughly

**Table 1.** Quantitation of relative expression levels of transfected versus endogenous APC and Axin

**Summary Axin overexpression**

| Experiment | I | II | III |
|---|---|---|---|
| Ratio GFP-FlyAxin to GFP-hAxin1 | 1.04 | 0.11 | 0.48 |
| Ratio GFP-hAxin1 to endo hAxin1 | 33.87 | 431.67 | 62.82 |
| Transfection efficiency | 30% | 42% | 36% |
| Fold overexpression level of GFP-FlyAxin to endo hAxin1 | 117.41 | 113.06 | 83.76 |

**Summary APC2 overexpresssion**

| Experiment | I | II | III |
|---|---|---|---|
| Ratio Flag-FlyAPC2 to Flag-hAPC1-1338 | 17.31 | 138.45 | 28.02 |
| Ratio Flag-hAPC1-1338 to endo hAPC1-1338 | 25.17 | 5.86 | 14.69 |
| Transfection efficiency | 30% | 33% | 25% |
| Fold overexpression level of Flag-FlyAPC2 to endo hAPC1-1338 | 1452.30 | 2458.54 | 1646.46 |

1400–2400 (*Table 1*). We suspect this is an over-estimate of the relative ratio of *Drosophila* APC2 to normal levels of hAPC1 in a wild-type colon cell, as the truncated APC1 protein present in SW480 cells would accumulate at lower levels than wild-type APC1 if it is subjected to nonsense-mediated mRNA decay, like many other proteins with early stop codons and like other truncated *APC1* alleles (*Castellsagué et al., 2010*; *Popp and Maquat, 2013*). Further, we also may need to reduce the ratio by a further factor of two as it is probable SW480 cells carry only one copy of the truncated APC allele, as most colorectal tumors either have the second allele mutated early enough to not produce a truncated protein or have lost the second allele by deletion (*Christie et al., 2013*). Regardless, it is important to remember that while the puncta provide a useful and visualizable model of the destruction complex, we are examining over-expressed proteins.

## A novel Axin:APC association site regulated by APC motifs R2 and B

To probe destruction complex structure and dynamics, we first need to define mechanisms of complex assembly. In the current model, destruction complex assembly is mediated by interaction of APC2's SAMPs and Axin's RGS domain (*Fagotto et al., 1999*; *Figure 1A*). Consistent with this, deleting the SAMPs disrupted APC2 recruitment into Axin puncta. However a shorter APC2 mutant, lacking both the SAMPs and R2 and B and thus resembling the truncated hAPC1 in tumors (*Figure 1B*), did co-localize with Axin (*Roberts et al., 2011*). We further found that deleting R2 and/or B, both of which are essential for APC function in flies and mammals, restored colocalization of Axin and APC2ΔSAMPs (*Roberts et al., 2011*). Thus a second means of mediating APC2:Axin interactions must exist, that is separate from the known SAMP:RGS interaction.

To identify the protein sequences mediating this putative R2/B regulated mechanism by which APC2 and Axin can interact, we truncated APC2 from its C-terminus, and assessed APC2 mutant colocalization with Axin. APC2's Arm rpts (*Figure 1B*) formed cytoplasmic puncta when expressed alone (*Figure 1—figure supplement 2A,B*), consistent with APC2's ability to self-oligimerize (*Kunttas-Tatli et al., 2014*). When co-expressed with Axin, the Arm rpts and Axin puncta associate, suggesting the Arm rpts are the Axin association site (*Figure 1C,D* arrows; intriguingly they often did not precisely colocalize, in contrast with the full length proteins (*Roberts et al., 2011*)). Next we defined where on Axin the Arm rpts associate, by expressing Axin fragments and conducting co-immunoprecipitation (co-IP; *Figure 1E,F*). Both full length Axin and a region including the GSK3 binding site co-IPed with APC's Arm rpts (*Figure 1E,F*). Axin's DIX domain was weakly detected in co-IPs, but this may occur by polymerization with full-length endogenous human Axin1 (hAxin1). Thus fly APC2's Arm rpts can mediate association with Axin's middle region. This was exciting, as it may explain the Arm rpts essential role in Wnt regulation (*McCartney et al., 2006*; *Roberts et al., 2012*).

To determine if this second APC:Axin interaction mechanism is conserved in humans and whether it occurs between proteins expressed at endogenous levels, we used SW480 cells, in which endogenous hAPC1 is truncated after 20R1 at 1338aa, thus lacking the SAMPs, R2 and B (*Figure 1B*). hAPC1-1338aa co-IPed with endogenous hAxin1 (*Figure 1G*). To verify that association was through the Arm rpts, we tested whether hAPC1's Arm rpts alone (*Figure 1B*) colocalized with hAxin1 in SW480 cells. hAPC1Arm rpts and hAxin1 each formed cytoplasmic puncta (*Figure 1—figure supplement 2C–E*). When coexpressed, they colocalized in cytoplasmic puncta (*Figure 1H,I* arrows), suggesting both hAPC1 and fly APC2 can associate with the Axin complex in two ways: the known interaction via the SAMPs and this novel interaction via the Arm rpts.

## Axin and APC2 form structured macromolecular complexes in vivo

Given this new data on destruction complex assembly, we next explored the structures assembled by Axin vs Axin plus APC, using the puncta formed in SW480 cells as a visualizable destruction complex model. While current data suggest Axin polymerization and APC recruitment are essential for destruction complex function, previous microscopy provided only limited information about the internal structure of Axin–APC complexes in vivo. In these images, Axin and APC colocalized in puncta without resolvable internal structure (e.g., *Mendoza-Topaz et al., 2011*; *Roberts et al., 2011*; *Figure 2A,B*). Recent advances increased the resolution possible with light microscopy. We thus used the superresolution approach Structured Illumination Microscopy (SIM) to visualize APC:Axin complexes, revealing a striking and previously undescribed internal structure. Cytoplasmic complexes formed by Axin alone appear to consist of Axin cables/sheets, assembling into hollow structures (*Figure 2C–F*; *Videos 1–3* show 3D reconstructions of these three puncta). Interestingly, coexpressing APC2 substantially altered the structure of many complexes, increasing the length/complexity of Axin cables (*Figure 2G, H–J* = three representative puncta; *Videos 4–6* show 3D reconstructions), with APC2 and Axin cables intertwined and in some puncta APC bridging Axin cables (*Figure 2H–J*, arrows).

These data suggest APC2 may enhance assembly of Axin oligomers, stimulating formation of larger complexes. To test this hypothesis, we quantified puncta cross-sectional area in Axin vs APC + Axin expressing cells, making z-projections of transfected cells and using the ImageJ particle analyzer. This revealed APC2:Axin puncta average almost twice the cross-sectional area of puncta formed by Axin alone (*Figure 2K*, left). We confirmed this by comparing the volume of Axin and APC:Axin complexes, as measured from our SIM data—once again APC + Axin puncta were on average significantly larger than those assembled from Axin alone, and the largest APC + Axin puncta were larger than any of the Axin puncta (*Figure 2L*). When we compared the average number of complexes in Axin alone vs APC2 + Axin expressing cells, we found twice as many complexes in cells expressing Axin alone (*Figure 2K*, right), suggesting that in the absence of functional APC, Axin forms more numerous but smaller puncta. These data support the hypothesis that APC2 induces changes in the structure of Axin complexes formed in vivo, enhancing Axin assembly into higher order complexes.

## APC2 stabilizes the Axin complex and promotes destruction complex throughput of βcat

The effect of APC on destruction complex structure might suggest APC helps regulate Axin turnover within the destruction complex. Previous analyses suggested that Axin dynamics within the destruction complex puncta correlate with function. Axin puncta are dynamic protein assemblies, and the Wnt effector Dishevelled significantly increases Axin dynamics in puncta, suggesting it negatively regulates Axin self-association (*Schwarz-Romond et al., 2007*), consistent with its known negative regulatory role in destruction complex function. To directly assess APC2 and Axin dynamics in the destruction complex and their influence on one another, we used Fluorescent Recovery After Photobleaching (FRAP) to measure dynamics of both APC2-GFP and Axin-RFP (*Figure 3A–D*). We assessed both recovery (mobile) fraction and $t_{1/2}$, using unbleached puncta as controls. Recovery fraction assesses what percentage of molecules in a complex turnover in the experimental time frame, and $t_{1/2}$ reflects the rate at which the dynamic fraction is exchanged—they are not necessarily dependent on one another. We found APC2 is also a dynamic component of destruction complex puncta, reaching a recovery plateau of 40% and a $t_{1/2}$ of 150 s (*Figure 3B*); however, APC2 is not as dynamic as Axin (*Figure 3C*).

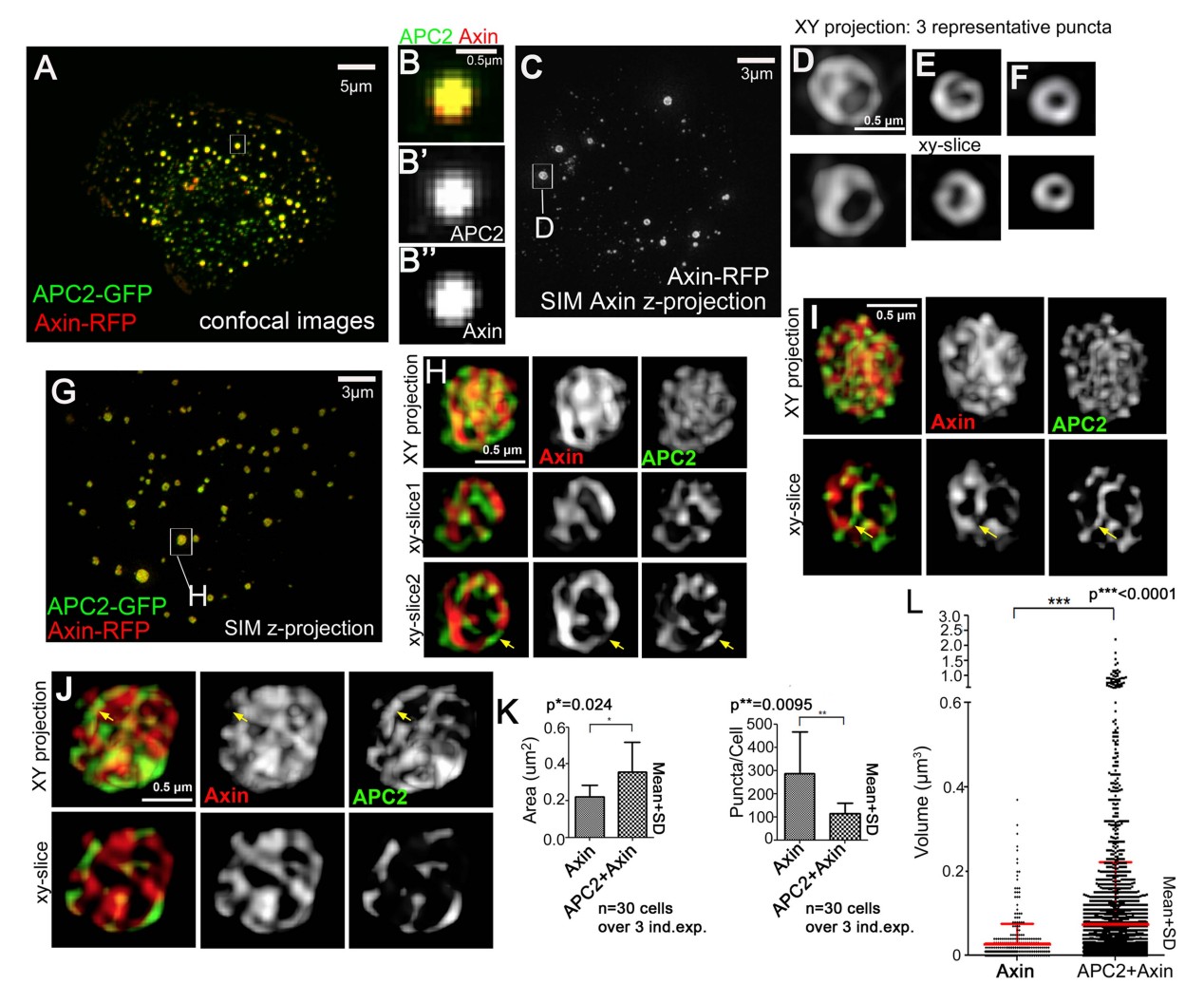

**Figure 2**. Axin and APC2 form structured macromolecular complexes in vivo. SW480 cells. (**A**) Confocal image, GFP-APC2 and Axin-RFP. APC2 is recruited into Axin puncta. (**B**) Closeups, showing failure to resolve internal structure. (**C–J**) SIM super-resolution. (**C**) Axin-RFP alone. (**D–F**) Closeups, Axin complexes from different cells. D = punctum boxed in **C**. Axin cables assemble into spheres/sheets. (**G**) Cell coexpressing GFP-APC2 and Axin-RFP. (**H–J**) Closeups of APC2:Axin complexes from different cells. H = punctum boxed in **G**. Axin cables increase in complexity and APC2 forms cables intertwined with Axin (arrows). (**K**) Analysis of confocal images. Complexes formed by APC2 and Axin average nearly twice the cross-sectional area of complexes formed by Axin alone (left). Axin-expressing cells have twice as many complexes as cells coexpressing APC2 + Axin (right). Student's *t*-test. (**L**) Puncta volume in Axin expressing cells (n = 3) vs APC2 + Axin expressing cells (n = 11) showing volumes across puncta population. Volume differences expressing cells are consistent with area quantification in (**K**). ANOVA-Bonferroni was used.

To test the hypothesis that APC stabilizes assembly of Axin monomers into the multimeric destruction complex, we compared Axin dynamics in puncta containing Axin alone with those containing Axin plus APC2. Interestingly, Axin expressed alone was quite dynamic, with a recovery plateau of almost 90% and a $t_{1/2}$ = 150 s (*Figure 3C,E*). However, when Axin was coexpressed with APC2, Axin dynamics were significantly reduced (recovery plateau = 40% and $t_{1/2}$ = 300 s; *Figure 3D,E*), suggesting APC stabilizes Axin assembly within puncta.

Axin cannot target βcat for destruction in APC's complete absence, even when Axin is overexpressed (*Mendoza-Topaz et al., 2011*). However, in SW480 cells, which express both truncated hAPC1 and endogenous hAPC2 (*Maher et al., 2009*), Axin overexpression can increase βcat destruction (*Nakamura et al., 1998*). If APC's role in the active destruction complex is to stabilize Axin assembly and thus the destruction complex scaffold, then APC should enhance Axin's ability to

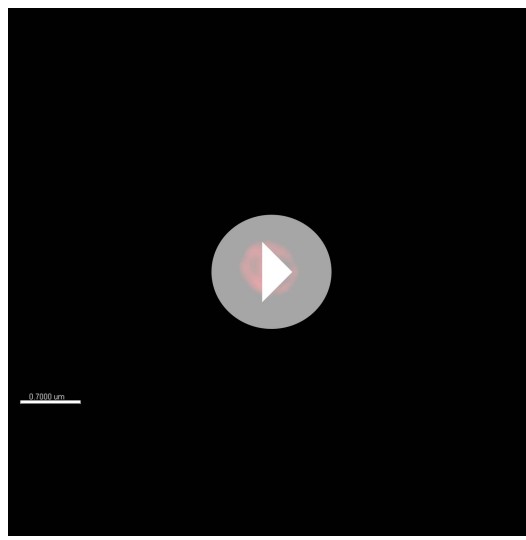

**Video 1.** 3D reconstruction of SIM superresolution image of Axin-RFP expressed in SW480 cells (see also *Figure 2D*). Volume view from Imaris 5.5 was used for reconstruction.

target βcat for destruction. We thus examined whether Axin alone fully restores βcat destruction, or whether adding APC2, either to endogenous wild-type Axin or co-expressed with fly Axin, further facilitates this. We began by measuring βcat fluorescence intensity in z-projections of cells transfected with either APC2 + Axin or Axin alone, using untransfected cells as internal controls (*Figure 3F–H*). Interestingly, while Axin reduced total βcat levels (*Figure 3F,G*), βcat was further reduced in cells expressing both APC2 and Axin (*Figure 3F,I*) suggesting that APC2 promotes more effective destruction complex activity.

Cells expressing APC2 'alone' also had strong βcat reduction (*Figure 3F,H*), presumably due to interaction with endogenous human Axin. We thus tested whether fly APC2 can and does interact with human Axin. Fly GFP-APC2 colocalizes in puncta with exogenous hAxin1-RFP (*Figure 3J*), and more importantly, endogenous hAxin1 coIPs with Flag-APC2 expressed in SW480 cells (*Figure 3K*). One further caveat is that we were assessing the ability of APC2 and Axin to promote βcat destruction after over-expression. To determine if differing levels of over-expression might explain the differences between Axin and APC2 + Axin (or 'APC2 alone'), we quantitated level of Axin or APC2 expression in a given cell by measuring levels of GFP/RFP fluorescence, and in parallel assessed levels of βcat in that cell (via fluorescence intensity). Strikingly, βcat levels were more effectively reduced by APC2 + Axin or by 'APC2 alone' than by Axin at all levels of expression assessed. As noted above, in all cases, ability to reduce βcat levels was somewhat diminished at the highest levels of expression (*Figure 1—figure supplement 1B*)—this may be because at very high expression levels, the transfected protein forms non-functional complexes with only a subset of the destruction complex proteins, as was previously suggested (*Lee et al., 2003*). Together these data support the hypothesis that APC2 enhances Axin's ability to promote βcat destruction.

Interestingly, in Axin-alone transfected cells, much of the excess βcat that accumulated (*Figure 3F*) was in Axin puncta (*Figure 3G'*, inset, arrow). In contrast, puncta in cells co-expressing APC2 and Axin had almost undetectable βcat levels (*Figure 3I'*, inset). We thus hypothesized that APC enhances Axin's ability to promote βcat exit from the destruction complex, and thus the destruction complex's βcat throughput. To further explore this, we examined phospho-βcat levels. SW480 cells, like other colon cancer cell lines with hAPC1 truncated before the Mutation Cluster Region (MCR; *Figure 1B*), have high phospho-βcat levels (*Yang et al., 2006*). This suggests that while Axin can facilitate βcat phosphorylation in these cells, Axin is less efficient at targeting βcat for destruction in the absence of wild-type APC1, and thus phosphorylated βcat accumulates. Strikingly, expressing APC2 alone or APC2 + Axin dramatically decreased phospho-serine 33/37 βcat (to ~20%

**Video 2.** 3D reconstruction of SIM superresolution image of Axin-RFP (see also *Figure 2E*).

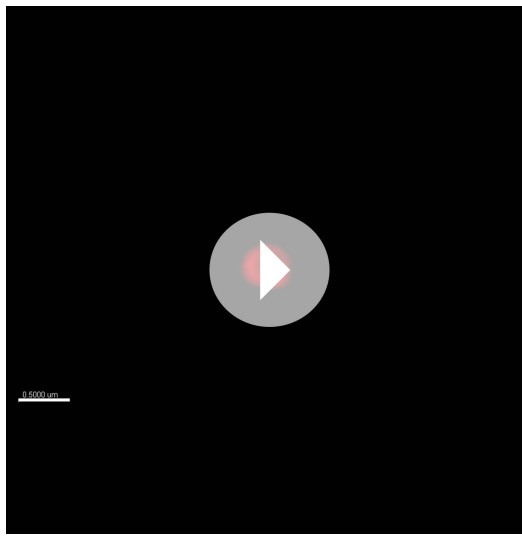

**Video 3.** 3D reconstruction of SIM superresolution image of Axin-RFP (see also *Figure 2F*).

that in untransfected cells), while Axin alone reduced phospho-βcat to only 60% (*Figure 3L,M*). These data further support the hypothesis that APC2 enhances Axin's ability to promote βcat destruction.

One potential caveat to the increased accumulation of βcat in puncta of cells expressing Axin alone vs those expressing APC2 plus Axin is that the former cells may simply have higher overall levels of βcat, thus resulting in higher accumulation in puncta. To address this, we inhibited βcat destruction using the proteasome inhibitor MG132. As others have previously observed (*Sadot et al., 2002*), proteasome inhibition elevates βcat levels. Proteasome inhibition elevates βcat levels both in cells expressing Axin alone and in those expressing APC2 plus Axin (*Figure 3—figure supplement 1C*). Strikingly, this allows it to accumulate in puncta even in cells expressing Axin + APC2 (*Figure 3—figure supplement 1A vs B*). However, cells expressing Axin still accumulate significantly higher levels of βcat than those expressing Axin plus APC2 (*Figure 3—figure supplement 1C*). Together, these data are consistent with a model in which APC increases βcat throughput of the destruction complex by stabilizing Axin assembly.

## Both APC2's Arm rpts and SAMPs are required to stabilize APC2:Axin complexes and regulate its substructure

Since APC associates with Axin via two regions, the Arm rpts and SAMP motifs (*Figure 1*), we hypothesized each interaction helps stabilize destruction complex assembly. To test this, we first

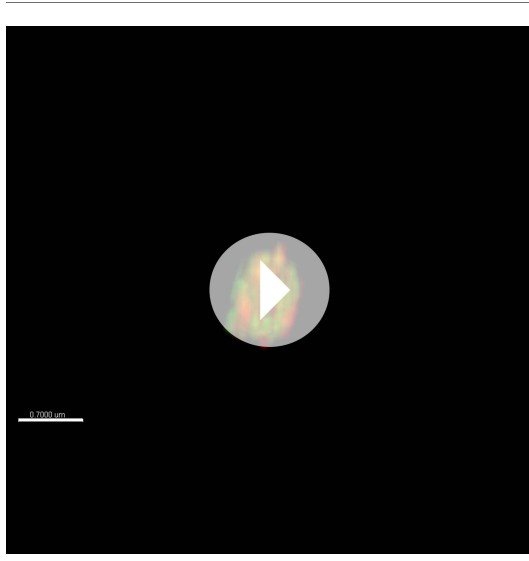

**Video 4.** 3D reconstruction of SIM superresolution image of GFP-APC2 and Axin-RFP expressed in SW480 cells (see also *Figure 2H*). Volume view from Imaris 5.5 was used for reconstruction.

measured APC2 dynamics when either the Arm rpts or SAMPs were individually deleted (*Figure 4A*). Deleting either region increased APC2 dynamics in Axin puncta; APC2ΔArm and APC2ΔSAMPs turnover reached higher plateaus (*Figure 4B*; 80–90% vs 40% for wild-type) in shorter times (*Figure 4B*; $t_{1/2}$ wildtype APC2 150 s; APC2ΔArm 75 s; APC2ΔSAMPs <25 s). Thus, APC2 needs both the Arm rpts and the SAMPs to stably associate with Axin complexes. Next we examined Axin dynamics in the presence of each APC2 mutant. Both the Arm rpts and SAMPs were required to stabilize Axin in destruction complexes, since Axin coexpressed with either APC2ΔArm or APC2ΔSAMPs exhibited the fast dynamics characteristic of Axin expressed alone (*Figure 4C*). Thus APC2 stabilizes APC:Axin complexes through multivalent interactions mediated by the Arm rpts and SAMPs.

Our SIM imaging suggested APC stabilization of Axin complexes altered their substructure. We thus tested whether both Axin interaction sites were essential for the effects on the structure of Axin puncta. APC2ΔArm, which retains the SAMPs, closely colocalized with Axin, even at

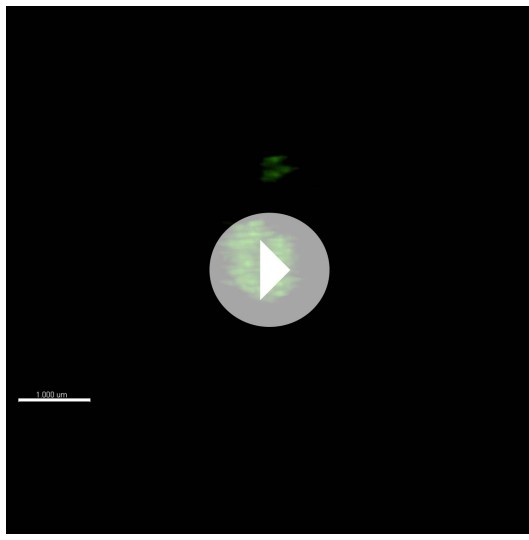

**Video 5.** 3D reconstruction of SIM superresolution image of GFP-APC2 and Axin-RFP (see also *Figure 2I*). DOI: 10.7554/eLife.08022.012

SIM resolution, rather than forming filaments of its own within puncta, like wild-type APC2 (*Figure 4D–F* vs *Figure 4H*). Further, Axin within these puncta remained simple in structure, similar to puncta formed by Axin alone (*Figure 4G*). APC2ΔSAMPs, which is much less tightly associated with Axin by either confocal localization or coIP (*Hart et al., 1998*; *Roberts et al., 2011*), did not strongly colocalize with Axin, instead forming a diffuse network surrounding Axin puncta. Importantly, APC2ΔSAMPs did not alter Axin structure within puncta as visualized by SIM. In the presence of APC2ΔSAMPs, Axin puncta retained the simpler structure of those formed by Axin (*Figure 4I–K*). Further, neither APC2ΔArm nor APC2ΔSAMPs increased Axin puncta size or reduced Axin puncta number (*Figure 4L–N*). Thus APC's ability to stabilize destruction complexes and stimulate growth of Axin cables requires both sites mediating Axin complex interaction, the Arm rpts and SAMPs. Both sites are also required to allow APC2 to efficiently downregulate βcat levels; APC2ΔSAMPs did not stimulate βcat destruction below Axin-alone mediated levels, while APC2ΔArmrpts could not downregulate βcat levels (*Figure 4O*). Since both APC2's Arm rpts and the SAMPs are essential for Wnt regulation in *Drosophila* (*Roberts et al., 2011*, *2012*), this suggests that APC2's ability to stabilize destruction complex assembly through its multivalent interactions is critical for destruction complex throughput of βcat.

## R2 and B regulate APC2 dynamics and APC2's ability to enhance βcat throughput

Our earlier steady state analysis revealed that APC2 motifs R2 and B antagonized APC:Axin interaction when the SAMPs were deleted. Since both R2 and B are essential for APC function in targeting βcat for destruction in vivo, we extended our work to determine whether and how R2 and B affect the dynamics of APC:Axin interactions. Our data above reveal the new Axin complex-association site is in APC's Arm rpts (*Figure 1*). Since removing R2/B restored APC/Axin interaction even in the absence of the SAMPs (*Figure 5—figure supplement 1A–E*), we hypothesized R2/B negatively regulates APC2 Armrpt:Axin interaction and that release of this interaction is essential to allow phosphorylated βcat to be moved on to destruction.

Our hypothesis predicted deleting either R2 or B should stabilize APC2:Axin interaction, by enabling APC2's Arm rpts to also productively mediate association with Axin. To test this we assessed how deleting R2 or B (*Figure 5A*) affected APC2 dynamics by FRAP. Deleting either R2 or B had the predicted effect, decreasing APC2 turnover rate dramatically (*Figure 5B*), suggesting R2 and B regulate APC2 dynamics in

**Video 6.** 3D reconstruction of SIM superresolution image of GFP-APC2 and Axin-RFP (see also *Figure 2J*). DOI: 10.7554/eLife.08022.013

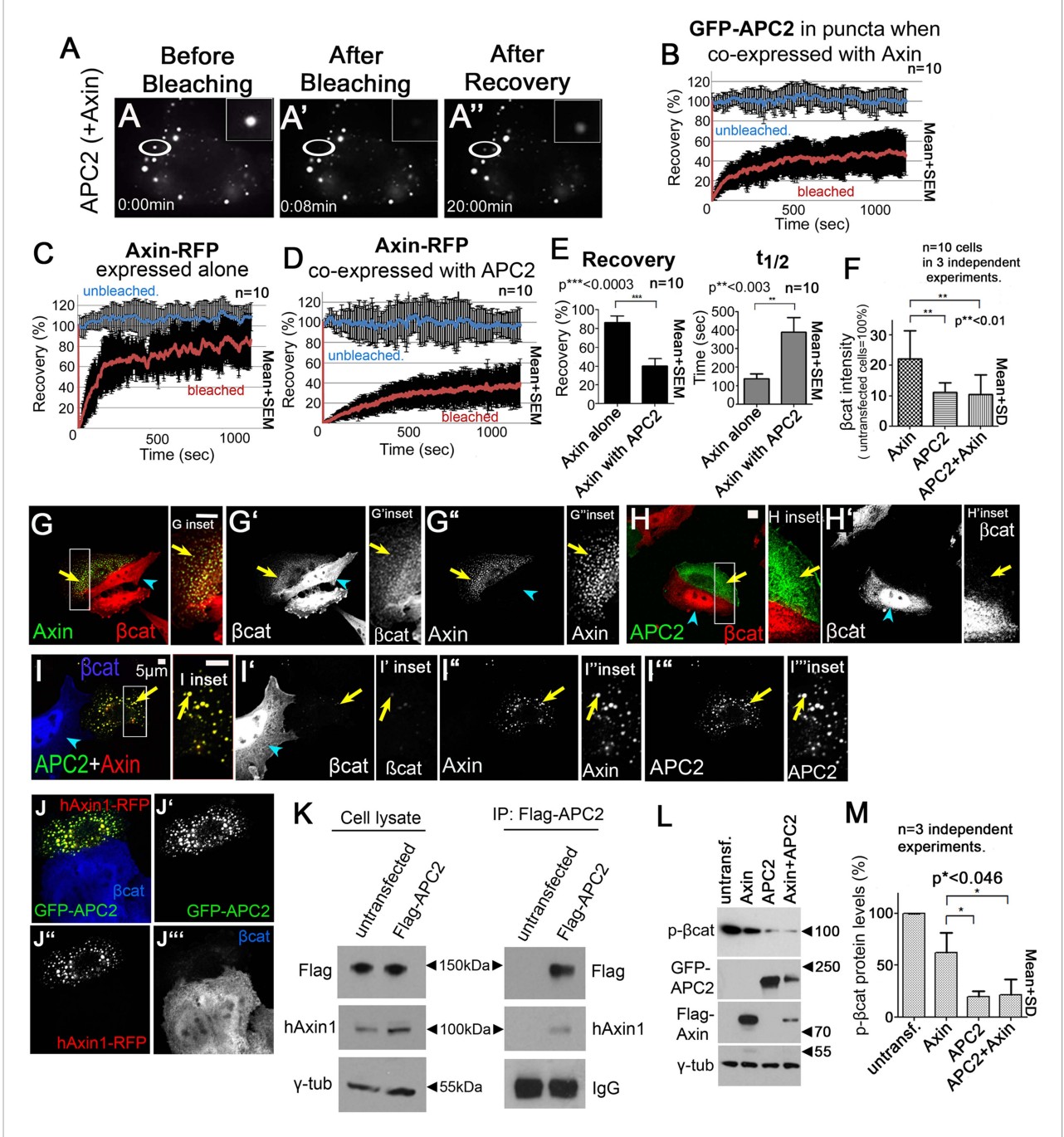

**Figure 3**. APC2 stabilizes Axin complexes and promotes efficient βcat destruction. (**A**) Stills, FRAP movie, SW480 cells transfected with GFP-APC2 (shown) and Axin-RFP. Inset = magnified APC2 signal in punctum. (**B**) APC2 recovers to ∼40% when in Axin puncta. Recovery curve (red); unbleached control (blue). (**C**) Axin expressed alone plateaus at ∼80%. (**D** and **E**) Axin is stabilized when coexpressed with APC2. (**F**) Total cell βcat fluorescent intensity normalized to untransfected cells (= 100%). APC2 or APC2 + Axin expression lead to stronger βcat reduction than Axin alone. (**G–I**) Indicated constructs expressed in SW480 cells. Insets = regions boxed. (**G**) GFP-Axin forms puncta and reduces βcat levels in this hAPC1 mutant cell line. βcat is detectable in puncta (arrows). (**H**) GFP-APC2 expressed alone is dispersed throughout the cell and βcat levels are low overall and in puncta. (**I**) Axin-RFP + GFP-APC2 coexpressed. βcat is reduced in APC2:Axin puncta (arrow) relative to puncta with Axin alone (**G**). (**J**) GFP-APC2 is recruited into puncta formed by human hAxin1-RFP. (**K**) Endogenous human hAxin1 co-IPs from SW480 cells with transfected Flag-APC2. Untransfected cells serve as a negative control. (**L** and **M**) Phospho-S33/37-βcat levels are more reduced when either APC2 or APC2 + Axin are expressed relative to Axin alone. (**L**) Immunoblot , transiently transfected SW480 cell extracts, centrifuged at 1000 rpm. (**M**) Quantification, phospho-S33/37-βcat protein levels from (**L**) and 2 replicates. Student's *t*-test.

*Figure 3. continued on next page*

*Figure 3. Continued*

The following figure supplement is available for figure 3:

**Figure supplement 1**. While proteasome inhibition reduces βcat destruction and causes βcat to detectably accumulate in APC2 + Axin puncta, it does not abolish the ability of APC2 to enhance Axin function in this regard.

the active destruction complex. While deleting R2 and B strongly stabilized APC2 in the complex, it did not significantly diminish APC2's ability to stabilize Axin, destruction complex size and structure (*Figure 5—figure supplement 1F–K*).

We next asked which step in the cycle of destruction complex function is blocked by removing R2 or B, and thus altering APC dynamics. To do so, we examined if βcat was retained in puncta containing these mutants. βcat was nearly undetectable when APC2 and Axin are coexpressed (*Figure 5C*, arrows). Interestingly, deleting either R2 or B led to βcat accumulation in APC2:Axin complexes (*Figure 5D,E*, arrows). Deleting R2 also abolished APC2's ability to enhance the reduction of βcat levels given by Axin alone (*Figure 5F*), suggesting that APC2 without R2 is not functional and therefore inhibited in its ability to promote βcat destruction via Axin complexes. βcat levels increased even further when motif B was deleted (*Figure 5F*) suggesting that APC2ΔB may interfere with βcat degradation via Axin. These data suggest that R2 and B regulate APC2 dynamics in the destruction complex and that APC2 lacking them cannot effectively support Axin in targeting βcat for destruction.

## R2 and B regulate APC2:Axin association via the Arm rpts

R2 or B thus regulate APC2:Axin interactions. We next determined which Axin association site, APC2's Arm rpts or SAMPs, was regulated. Deleting the SAMPs substantially reduced APC2 recruitment into Axin complexes while further deleting either R2 or B restored strong Axin:APC2ΔSAMPs colocalization (*Figure 5—figure supplement 1*); in APC2ΔSAMPs the only remaining means of interacting with the Axin complex was via the Arm rpts, suggesting Arm rpts:Axin association is regulated by R2/B. In contrast, deleting APC2's Arm rpts (*Figure 6—figure supplement 1A*) did not reduce Axin colocalization (*Figure 6—figure supplement 1B* arrows); thus not surprisingly deleting either R2 or B in APC2ΔArm did not further alter this (*Figure 6—figure supplement 1A,C,D* arrows).

To directly test the hypothesis that R2 and B regulate dynamics of Arm rpts:Axin association, we returned to our FRAP assay. Consistent with effects on colocalization, deleting R2 or B (*Figure 6A*) decreased APC2ΔSAMPs dynamics (*Figure 6B*; $t_{1/2}$ increased significantly for both mutants; intriguingly recovery fraction was only significantly affected by deleting R2, perhaps reflecting the unaltered destruction complex structure in these mutants). Co-IPs extended the FRAP results, revealing more stable APC2ΔSAMPs:Axin interaction when R2 was deleted (*Figure 6C*; quantified in *Figure 6D*; the change after deleting motif B and the SAMPs did not reach statistical significance). Thus without R2, APC2's Arm rpts associate more robustly with Axin complexes, slowing APC2 dynamics. In contrast, SAMPs:Axin interaction was not regulated by R2 or B: APC2ΔArm (in which the only remaining interaction with Axin was via the SAMPs), APC2ΔArmΔR2, and APC2ΔArmΔB all had similar recovery plateaus and $t_{1/2}$ (*Figure 6—figure supplement 1E*), and co-IPs showed no difference in APC2:Axin association among these constructs (*Figure 6—figure supplement 1F,G*). In fact, APC2ΔR2 and APC2ΔB, which retain both the Arm rpts and SAMPs, coIPed with Axin as or more robustly than wild-type APC2 (*Figure 6—figure supplement 1H,I*) consistent with enhanced APC:Axin interaction through increased Arm rpts:Axin association when R2 and B were deleted. Thus APC2 and Axin have two distinct interaction interfaces with different properties: strong association via the SAMPs is independent of R2/B and a second interaction via APC2's Arm rpts is controlled by R2/B.

## APC2:Axin association via the Arm repeats is regulated by GSK3 kinase

The known essential role of R2 and B in the active destruction complex in targeting βcat for destruction suggested their ability to regulate APC:Axin interaction is critical. We thus further explored the mechanism by which they regulate destruction complex dynamics and function, by comparing sequences of these adjacent motifs in mammalian and fly APCs (*Figure 7A*). Strikingly, threonines and serines were among the most highly conserved residues. In the destruction complex, GSK3 and CK1 phosphorylate not only βcat but also Axin and other sites on APC (*Ikeda et al., 1998*; *Yamamoto et al., 1999*; *Ha et al., 2004*).

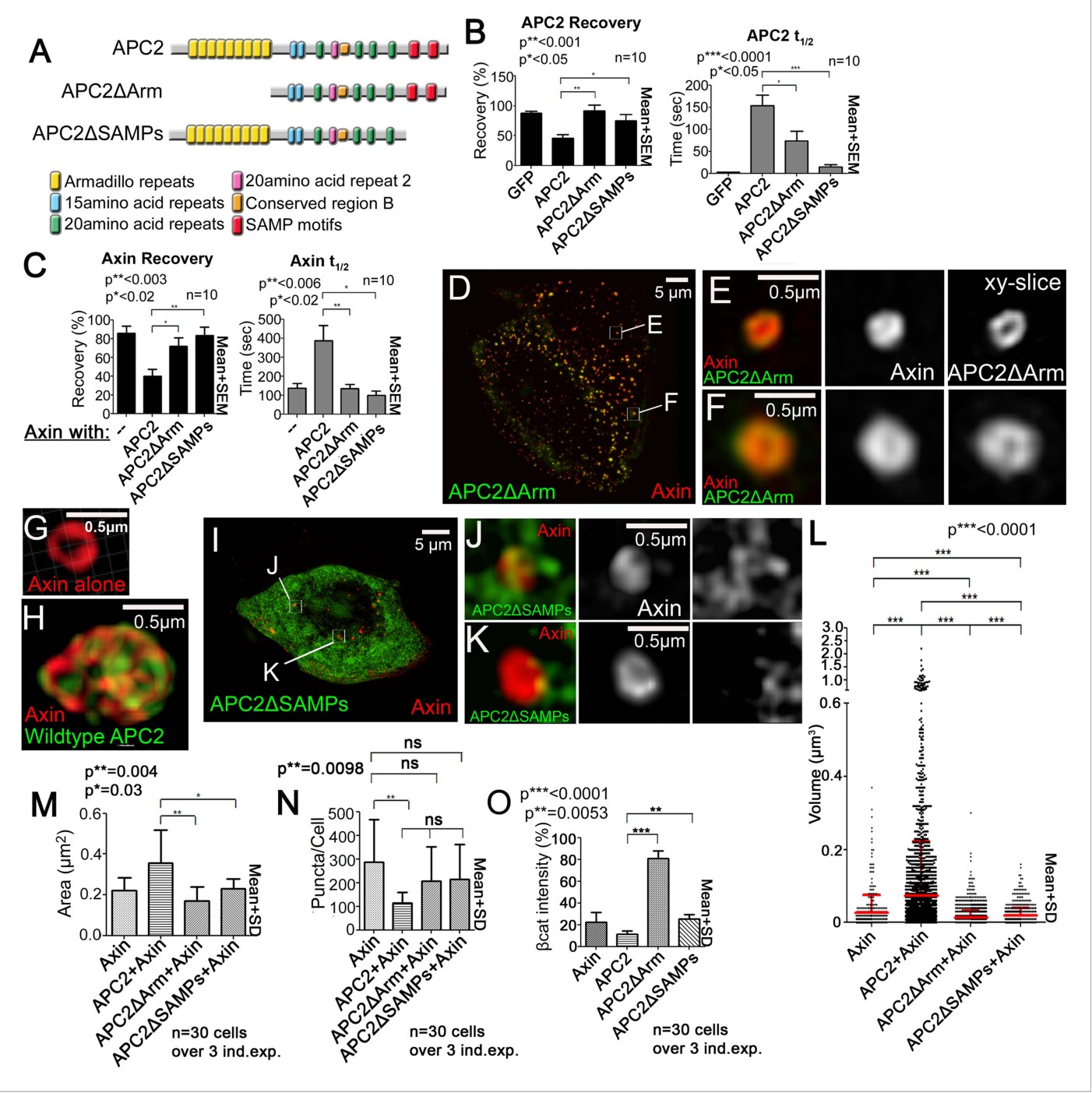

**Figure 4**. APC2's Arm rpts and SAMPs each are required to stabilize APC2:Axin complexes. (**A**) APC2 mutants. (**B** and **C**) FRAP analyses, SW480 cells. (**B**) APC2 needs both the Arm rpts and SAMPs to robustly associate with Axin puncta. Student's *t*-test. (**C**) Axin stabilization by APC2 is abolished when either APC2's Arm rpts or SAMPs are deleted. (**D,K**) SIM super-resolution images, SW480 cells expressing indicated constructs. (**D–F**) GFP-APC2ΔArm and Axin-RFP. (**E–F**) Close-ups, *X–Y* slice. Axin structure resembles complexes formed by Axin alone. (**G** and **H**) Axin-RFP puncta and APC2:Axin puncta for comparison. (**I–K**) GFP-APC2ΔSAMPs and Axin-RFP. (**J,K**) Close-ups. Axin does not form a complex internal structure when APC2 ΔSAMPs is expressed. (**L**) Puncta volume in SIM images of Axin (n = 3 cells), APC2 + Axin (n = 11), APC2ΔArm + Axin(n = 9), and APC2ΔSAMPs + Axin (n = 5) expressing cells. Deleting either the Arm rpts or the SAMPs inhibits APC2's ability to enhance puncta volume. ANOVA-Bonferroni. (**M**) Puncta area, confocal images. Area differences are consistent with volumes in (**L**). Student's *t*-test. (**N**) Puncta number, confocal images. Deleting Arm rpts or SAMPs in APC2 fails to decrease number of APC2:Axin puncta as does wildtype APC2. (**O**) APC mutants lacking the Arm rpts or SAMP motif show decreased ability to reduce βcat levels in SW480 cells. Quantification, total cell βcat fluorescent intensity normalized to untransfected cells.

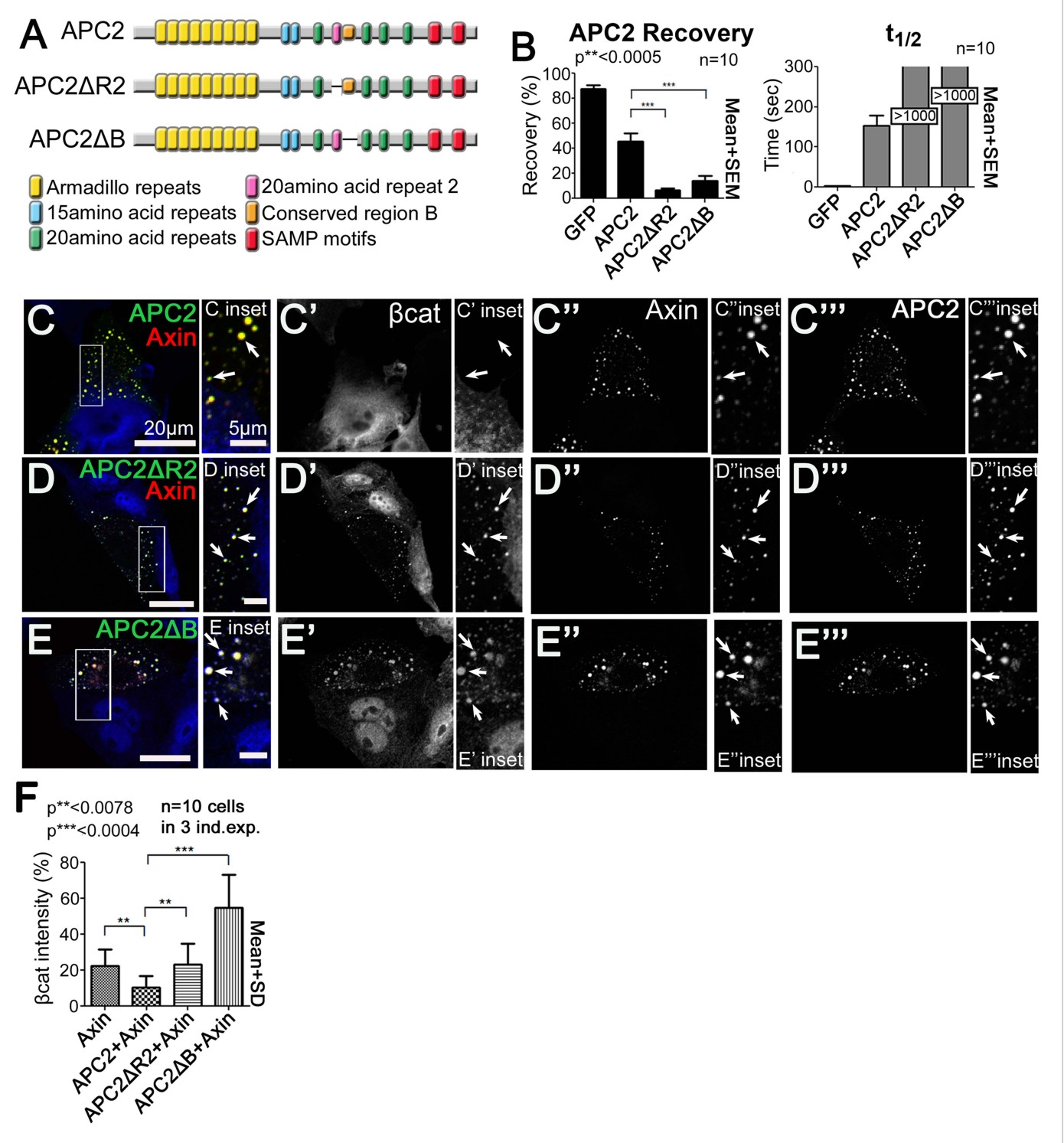

**Figure 5**. R2 and B regulate APC2 dynamics in the destruction complex, and regulate βcat removal from the destruction complex. (**A**) APC2 mutants. (**B–F**) SW480 cells transfected with Axin-RFP and indicated GFP-APC2 constructs (**B**) FRAP assay. Deleting either R2 or B slows APC2's turnover in Axin puncta. (**C–E**) Axin-RFP, GFP-APC2 constructs, βcat (inset = boxes). (**C**) βcat is essentially undetectable in APC2:Axin puncta (arrows). (**D**) βcat accumulates in APC2ΔR2:Axin puncta (arrows). (**E**) βcat strongly accumulates in APC2ΔB:Axin puncta (arrows). (**F**) Deletion of R2 or B impair the ability of APC2 to aid Axin in reducing βcat fluorescent intensity.

*Figure 5. continued on next page*

*Figure 5. Continued*

The following figure supplement is available for figure 5:

**Figure supplement 1**. Colocalization of APC2's Arm repeat domain with Axin is controlled by R2 and B and APC2 without R2 or region B still stabilizes Axin complexes.

Motif B has multiple serines matching GSK3's phosphorylation consensus and one match to the CK1 consensus, while R2 has several matches to both GSK3 and CK1 consensuses (*Figure 7A*). GSK3 kinase plays multiple roles in promoting destruction complex activity, phosphorylating βcat to target it to the E3 ligase and also regulating the destruction complex by phosphorylating the βcat binding sites on APC, increasing their affinity, and phosphorylating Axin. We hypothesized GSK3 also phosphorylates R2 and/or B, to trigger release of APC2's Arm rpts from Axin.

To test this we first determined whether blocking GSK3 activity using LiCl, a well-known GSK3 inhibitor (*Klein and Melton, 1996*), or BIO, a very specific GSK3 inhibitor (*Meijer et al., 2003*) affected APC2ΔSAMPs:Axin interaction (we verified GSK3 inhibition by assessing βcat accumulation in APC2:Axin puncta; *Figure 7—figure supplement 1A–F*; *Stambolic et al., 1996*). APC2ΔSAMPs (*Figure 7B*) is only weakly recruited into Axin puncta (*Figure 7C*). If GSK3 phosphorylation of R2/B antagonizes Arm rpts:Axin association, inhibiting GSK3 should increase colocalization of APC2ΔSAMPs with Axin, as did deleting R2 or B (*Figure 5—figure supplement 1*). Strikingly, while APC2ΔSAMPs is

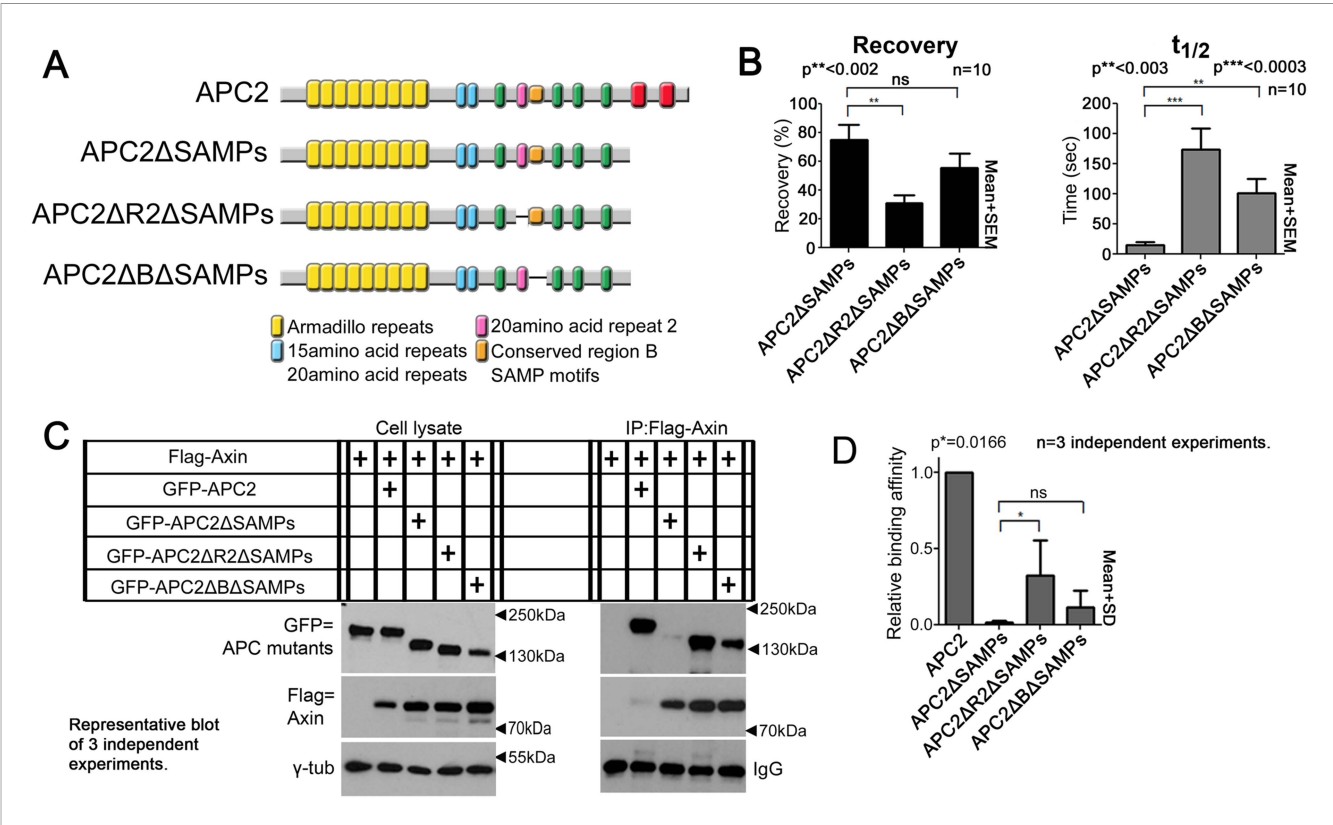

**Figure 6**. Association of Axin with APC2's Arm rpts is controlled by R2 and B. (**A**) APC2 constructs. (**B**) Deleting either R2 or B in APC2ΔSAMPs slows APC2 recovery time and reduces recovery fraction. FRAP assay, SW480 cells transfected with GFP-tagged APC2 constructs and Axin-RFP. (**C**) IPs of indicated constructs. Deleting the SAMPs substantially decreases APC2:Axin coIP, but this is partially restored by deleting R2. (**D**) Quantification, >2 replicates, normalized to Axin pull down. Student's *t*-test.

The following figure supplement is available for figure 6:

**Figure supplement 1**. Binding of APC2's SAMP motif to Axin is not regulated by R2 and B.

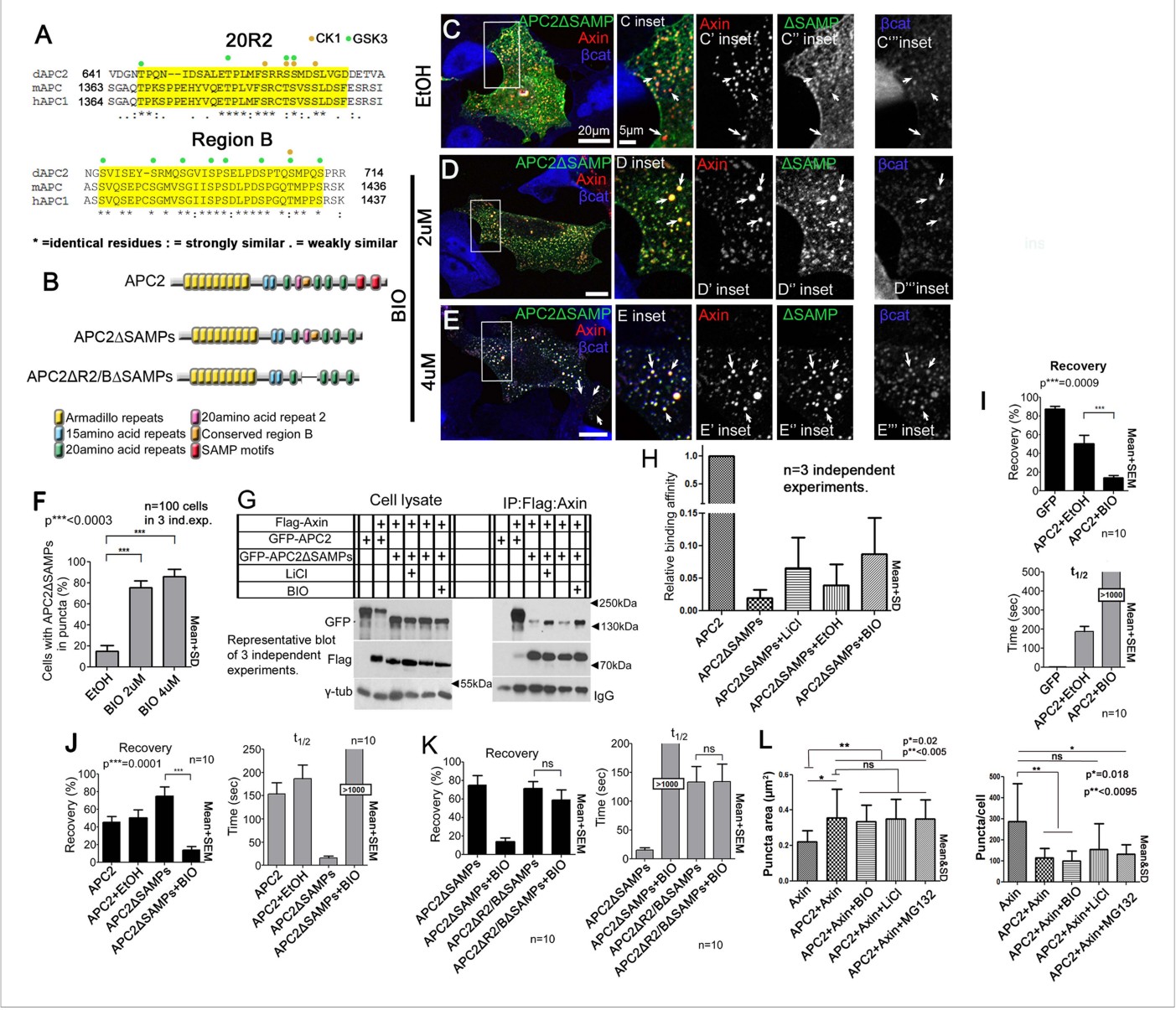

**Figure 7**. Axin:APC2 Arm rpts association is regulated by GSK3. (**A**) R2 and B of *Drosophila* dAPC2, mouse mAPC1 and human hAPC1. Potential CK1 (orange) and GSK3(green) phosphorylation sites. (**B**) APC2 mutants. (**C–E**) SW480 cells expressing GFP-APC2ΔSAMPs and Axin-RFP. Insets = boxes in **C–E** (**C**) Control (Ethanol treated (EtOH)). Deleting the SAMPs reduces APC2:Axin colocalization (arrows). (**D**) 2 µM BIO enhances APC2ΔSAMPs recruitment into Axin puncta. (**E**) Increasing BIO to 4 µm further boosts APC2ΔSAMPs recruitment into Axin puncta. (**F**) Quantification of (**C–E**). (**G**) CoIP of APC2ΔSAMPs with Axin in SW480 cells ± LiCl or BIO. Full length APC2 is a control. Deleting the SAMPs drastically reduces coIP but GSK inhibition partially restores this. (**H**) Quantification of coIP in G, >2 replicates, normalized to Axin. (**I**) FRAP assay, SW480 cells transfected with Axin-RFP + GFP-APC2. GSK3 inhibition decreases APC2 dynamics. (**J**) GSK inhibition also slows APC2ΔSAMPs dynamics. (**K**) Deleting R2/B in APC2ΔSAMPs abolishes effect of GSK3 inhibition on dynamics. Student's *t*-test. (**L**) GSK3 inhibition with either BIO or LiCl does not further increase the size or decrease the number of APC plus Axin puncta, nor does treatment with the proteasome inhibitor MG132. Puncta area and puncta number, from confocal images.

The following figure supplement is available for figure 7:

**Figure supplement 1**. GSK3 regulates association of APC2's Arm repeats with the Axin complex.

largely diffusely cytoplasmic (*Figure 7C*), BIO treatment strongly increased Axin:APC2ΔSAMPs colocalization (*Figure 7D* arrows), in a concentration dependent manner (*Figure 7E*). Consistent with this, only 16% of untreated control cells had APC2ΔSAMPs:Axin colocalization (*Figure 7F*), while

inhibiting GSK3 with BIO increased colocalization to 75% of cells (*Figure 7F*). LiCl also led to robust Axin:APC2ΔSAMPs colocalization (*Figure 7—figure supplement 1G* vs *Figure 7—figure supplement 1H*; quantified in *Figure 7—figure supplement 1I*). We used coIP to verify that APC2ΔSAMPs associates more robustly with Axin upon GSK3 inhibition. Deleting the SAMPs drastically reduced APC2:Axin coIP (*Figure 7G*; quantified in *Figure 7H*). GSK3 inhibition by either LiCl or BIO increased APC2ΔSAMPs coIP with Axin (*Figure 7G,H*). Thus GSK3 inhibition stabilizes steady state Axin: APC2ΔSAMPs association, as did deleting R2 or B, consistent with our model that phosphorylating these motifs normally antagonizes Axin:Arm rpts association.

The hypothesis that association of APC2's Arm rpts with Axin is regulated by GSK3 also predicts inhibiting GSK3 should affect APC2 dynamics in the destruction complex, as did deleting R2 or B. Consistent with this, inhibiting GSK3 dramatically decreased APC2's dynamics (*Figure 7I*; plateau reduced from 40% to 10%; $t_{1/2}$ increased from 150 to >1000 s). This suggests APC2 turnover in Axin complexes is regulated by GSK3. Our model further predicts that GSK3 regulates APC2 dynamics by regulating the APC2Arm rpts:Axin association. Thus GSK3 inhibition should stabilize APC2ΔSAMPs: Axin interactions and reduce APC2ΔSAMPs dynamics. As predicted, APC2ΔSAMPs rapid turnover was dramatically decreased by GSK3 inhibition (*Figure 7J*). In contrast, GSK3 inhibition had no effect on APC2ΔArm turnover (*Figure 7—figure supplement 1J*). These data suggest that GSK3 activity promotes release of APC2's Arm rpts from the Axin complex.

Our data are consistent with GSK3 acting via phosphorylating R2 and/or B, but could also be mediated via other effects of GSK3. To assess this, we examined whether inhibiting GSK3 affected turnover of an APC2ΔSAMPs mutant lacking both R2 and B (*Figure 7B*), reasoning this would assess the effect of GSK3 on regulation of the Axin:APC2Arm rpts interaction. Strikingly, the recovery plateau and $t_{1/2}$ of APC2ΔR2/BΔSAMPs were insensitive to GSK3 inhibition (*Figure 7K*), in contrast to APC2ΔSAMPs. These data are consistent with GSK3 affecting APC2 residence time in the destruction complex through R2/B. However, deleting R2/B and blocking GSK3 activity had different effects on APC2 recovery fraction, suggesting that not all effects of GSK3 are mediated through R2/B. GSK3 phosphorylates other targets in the destruction complex, and thus it is very likely GSK3 inhibition has additional means of altering destruction complex dynamics. Further, an unphosphorylated R2/B motif may affect APC's dynamics differently than deleting R2/B. Taken together, however, our data suggest that GSK3 acts in part through R2 and B to promote release of APC2's Arm rpts from the Axin complex.

If GSK3 inhibition stabilizes the interaction of APC2 and Axin, and as noted above, APC2 also can stabilize Axin in puncta, increasing their size (and thus decreasing puncta number), then GSK3 inhibition might synergize with APC, further increasing the size of Axin puncta. We thus inhibited GSK3 with either BIO or LiCl in cells co-expressing Axin and APC2 and examined both puncta size and number. GSK3 inhibition did not further increase puncta size or further decrease puncta number (*Figure 7L*). These data are consistent with our analysis above of APC2ΔR2 and APCΔB—these mutations also stabilize APC2 in the destruction complex (*Figure 5B*) but do not further increase puncta size or decrease puncta number (*Figure 5—figure supplement 1F–J*). We also tested whether proteasome inhibition might trigger enlargement of the APC + Axin complexes—this also did not further increase puncta size or further decrease puncta number (*Figure 7L*). Thus, while APC2 can stabilize Axin complexes, inhibition of GSK3 or the proteasome does not synergize with this.

## Mutating putative phosphorylation sites in B disrupts APC2's function in regulating βcat destruction in SW480 cells

We hypothesized that R2/B phosphorylation by GSK3 is a key step in APC2's mechanism to target βcat for destruction. Thus we tested whether R2/B can be phosphorylated by GSK3. A GST-fusion containing just R2 and B from either fly APC2 or human APC1 can be phosphorylated in vitro by human GSK3 (*Figure 8A*). Human R2/B was strongly phosphorylated, whereas fly R2/B was more weakly phosphorylated. Thus the potential phosphorylation sites in R2/B can be phosphorylated by GSK3, possibly at the GSK3 consensus sites (*Figure 8B*).

R2 and B are essential for APC2's function in βcat degradation. Our hypothesis suggests phosphorylation of R2/B promotes release of APC2's Arm rpts from the Axin complex, and that this would be essential for the catalytic cycle of the destruction complex—thus mutating these putative GSK3 phosphorylation sites in motif B would reduce APC2 function in helping mediate βcat

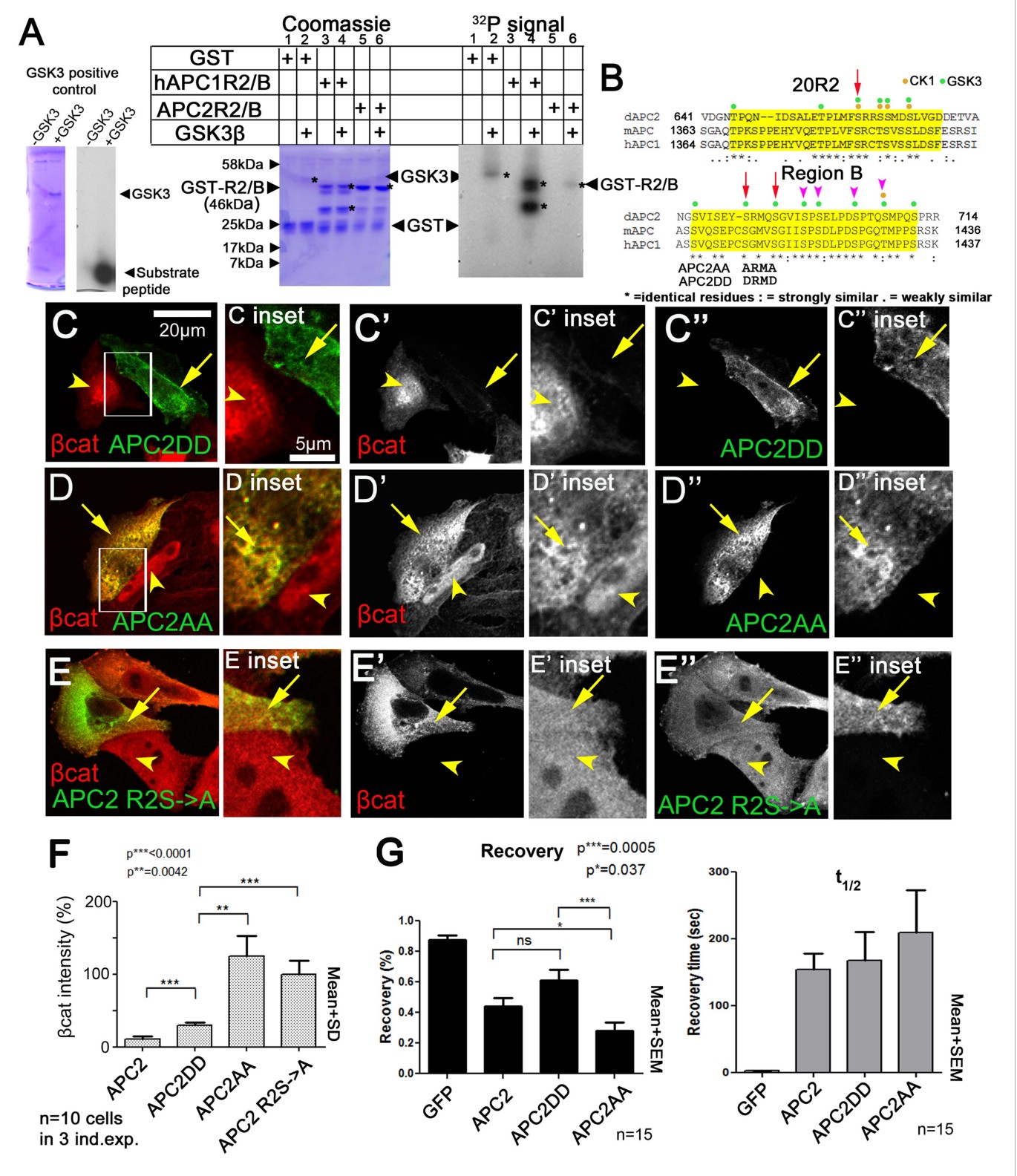

**Figure 8**. Mutating putative phosphorylation sites in B disrupts APC2 function. (**A**) R2/B of human or fly APCs can be phosphorylated by human GSK3. In vitro kinase assay, GSK3 substrate peptide (positive control, left panels), GST-tagged humanAPC1R2/B or fly APC2R2/B fragments. GST was a negative control. Left of each pair: Coomassie stained gel, right: Phosphorylation detected using $P^{32}$. Asterisks (*) indicate Coomassie-stained bands that align with $P^{32}$-labeled proteins. In the presence of GST alone, GSK3 autophosphorylates (lane 2). HumanAPC1R2/B is strongly phosphorylated (lane 4) while fly

*Figure 8. continued on next page*

*Figure 8. Continued*

APC2R2/B was more weakly phosphorylated (lane 6). Representative of two experiments. (**B**) R2 and B of *Drosophila* dAPC2, mouse mAPC1 and human hAPC1. Potential CK1 (orange) and GSK3 (green) phosphorylation sites. Red Arrow in R2 = serine mutated to alanine in mutant assessed in panel **E**. Red arrows in B = serines mutated to aspartic acid or alanine in APC2AA or APC2DD. Magenta arrowheads = additional serines mutated in 4 serine and six serine mutations (data not shown). (**C**) APC2DD (2 serines in B changed to aspartic acid; arrows in **B**) effectively reduces βcat levels in SW480 cells (arrow vs arrowhead). Inset = box in **A**. (**D**) APC2AA (2 serines in B changed to alanine, arrows in **B**) is unable to target βcat for destruction (arrow vs arrowhead). (**E**) APC2 R2S->A (single serine in R2 changed to alanine, arrow in **B**) is unable to target βcat for destruction (arrow vs arrowhead). (**F**) Quantification, total βcat fluorescent intensity. Student's *t*-test. (**G**) FRAP, Axin-RFP + GFP-APC2 constructs. APC2AA reaches a lower recovery plateau than either wild-type APC or APC2DD.

destruction. To begin to test this, we replaced 2 (APC2AA; *Figure 8B*, red arrows), 4 or 6 (*Figure 8B*, magenta arrowheads) conserved serines in B that match the GSK3 consensus with alanine, to prevent phosphorylation. All reduced function in downregulating βcat levels (see below and data not shown). We thus focused on the least altered of these, the mutant that replaced the more N-terminal two serine residues with alanine (APC2AA; *Figure 8B*, red arrows), thus preventing phosphorylation. We also created a mutant that replaced these same two residues with aspartic acid, creating a phosphomimetic APC2 (APC2DD). Strikingly, while APC2DD effectively reduced βcat levels (*Figure 8C,F*), APC2AA was unable to do so. APC2AA cells accumulated βcat at levels as high or higher than adjacent untransfected cells (*Figure 8D,F*), suggesting these two amino acid missense mutations substantially reduced APC2 function. APC2DD substantially reduced βcat levels (to ~30%), but was not quite as effective as wildtype APC2 (*Figure 8F*). This statistically significant difference may suggest dephosphorylation of these residues is also required for full APC2 function. We also found mutating a single conserved serine residue in R2 to alanine (APC2 R2S->A (=APC2S660A); *Figure 8B*) strongly diminished APC2 function in reducing βcat levels (*Figure 8E,F*). Together, these data are consistent with a model in which phosphorylation of conserved serines in APC2 motifs R2 and B are important for APC's function in the destruction complex to target βcat for degradation. Finally, we tested whether the first of these putative phosphomutants affected APC2 dynamics in the FRAP assay. Since GSK3 inhibition slowed APC's dynamics, we predicted APC2AA should have a lower turnover rate than wildtype APC or APC2DD. We saw a subtle but statistically significant reduction in APC2AA recovery fraction (*Figure 8G*). However this was not nearly as dramatic as that of deleting R2 or B (*Figure 5B*); perhaps this due to the fact that we only altered two of several potential phosphorylation sites. Together these data are consistent with the idea that phosphorylation of conserved serine residues in R2/B regulates APC's function in the destruction complex, but since the effect on dynamics was substantially less dramatic than that of deleting R2 or B, it suggests other residues in R2 and B may also contribute to regulating APC2 dynamics. Further, it is clear that GSK3 has other effects on the complex, complicating interpretation of its inhibition. It will be important to examine this in the future.

## Mutating putative phosphorylation sites in B severely reduces APC2's function in regulating βcat destruction or cell fate in vivo in *Drosophila*

To test the role of these two putative phosphorylation sites in APC function in an in vivo context where we can examine both cells receiving and not receiving Wnt signals, we turned to *Drosophila*, where we can express mutant APC2 under control of the endogenous *APC2* promoter in the complete absence of all endogenous APC function, using embryos maternally and zygotically *APC2 APC1* double mutant. We generated transgenics expressing APC2DD or APC2AA using the endogenous APC2 promotor (as in *Roberts et al., 2011*), and crossed them into the *APC2 APC1* mutant background. All progeny were maternally *APC2 APC1* mutant and 50% of progeny were also zygotically mutant, while the other 50% were paternally rescued (we cannot generate homozygous double mutant males). In the absence of the transgene, 43% of progeny hatch (*Figure 9A*), consistent with ~50% zygotic rescue. A wild-type APC2 transgene expressed in *APC2 APC1* mutants led to 95% survival, comparable to wild-type flies (93–99%; *Figure 9A*). APC2DD was as effective as wild-type APC2 at restoring embryonic viability (96% survival; *Figure 9A*). In contrast, APC2AA expressed in *APC2 APC1* mutants only weakly rescued embryonic lethality (64% viability; *Figure 9A*).

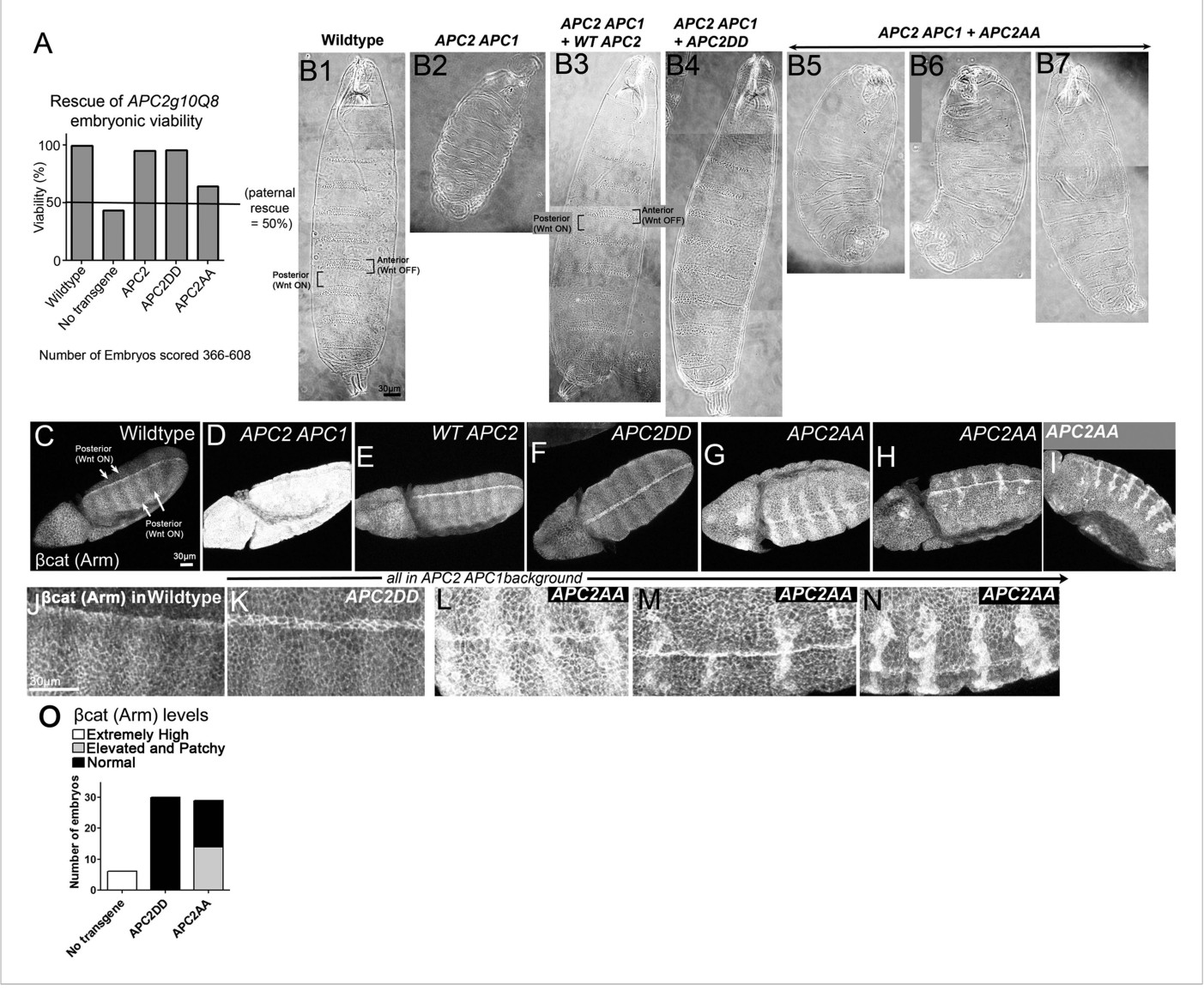

**Figure 9**. Blocking potential phosphorylation at 2 conserved serines in B disrupts APC2 function in the fly. APC2DD and APC2AA (*Figure 8B*) were expressed with the endogenous APC2 promoter in *APC2 APC1* maternal/zygotic double mutants. (**A**) *APC2 APC1* maternal/zygotic double mutants die as embryos (50% of embryos are zygotically rescued). APC2DD rescues embryonic viability as well as wildtype APC2. In contrast, APC2AA has only weak rescue ability. (**B**) Cuticles. (**B1**) Wildtype. Note pattern of anterior denticles (Wnt inactive) and posterior naked cuticle (Wnt active). (**B2**) Loss of APC2 and APC1 leads to denticle loss and expanded naked cuticle. (**B3**) Wildtype APC2 fully restores Wnt regulated cell fates of alternating denticles and naked cuticle. (**B4**) APC2DD similarly restores cell fates. (**B5–B7**) APC2AA largely fails to restore Wnt-regulated cell fates, and thus most cells secrete naked cuticle. Images = range of rescue ability. (**C–I**) βcat (fly Armadillo (Arm)) levels. Stage 9–10 embryos. (**J–N**) Close-ups of **C–I**. (**C,J**) Wildtype. Striped pattern of βcat indicative of Wg (fly Wnt) active and Wg inactive regions. (**D**) Loss of APC1 and APC2 leads to uniform very high levels of βcat. (**E**) WT APC2 restores normal βcat regulation, with higher levels in cells receiving Wg signal, and lower levels in other cells. However, Wg signal does not elevate βcat levels to those seen in embryos lacking functional APC. (**F and K**) APC2DD also rescues normal βcat regulation. (**G–I** and **L–N**) APC2AA restores some Wnt responsiveness, but βcat levels are elevated in all cells and especially elevated in a subset of cells receiving Wnt signal. (**O**) Quantification, embryos blind-scored.

We then examined rescue of cell fates. In wildtype embryos, a row of cells in each body segment expresses the Wnt homolog Wingless (Wg), thus regulating cell fate. In cells not receiving Wg signal, the destruction complex effectively destroys βcat that is not sequestered in cadherin-catenin complexes at the cell membrane, and cells choose anterior fates and secrete cuticle covered with denticles (*Figure 9B1*). In contrast, cells receiving Wg signal accumulate cytoplasmic and nuclear βcat, choose posterior fates, and secrete naked cuticle (*Figure 9B1*). Maternal/zygotic *APC2 APC1* mutants

cannot destroy βcat and thus all cells accumulate extremely high levels of βcat, resulting in a smaller embryo in which all surviving cells choose posterior fates and secrete naked cuticle (*Figure 9B2*; *Ahmed et al., 2002*; *Akong et al., 2002*). Like our wildtype APC transgene (*Figure 9B3*), APC2DD fully restored alternating anterior (denticle) and posterior (naked cuticle) fates, resembling wild-type (*Figure 9B4*). In contrast, APC2AA largely failed to restore cell fates. Most embryos lost nearly all denticles (*Figure 9B5–7*), thus resembling embryos expressing an indestructible form of βcat (*Pai et al., 1997*). This suggested APC2AA has severely reduced regulatory function, though it is not completely dead.

The final test was to examine how well these APC2 mutants restored βcat destruction. In wildtype (*Figure 9C,J*), Wg signal expressed in segmental stripes turns down destruction complex activity, leading to successive stripes of cells with only cortical βcat (destruction complex on) or with elevated cytoplasmic and nuclear βcat (destruction complex turned down). In contrast *APC2 APC1* maternal/zygotic mutants have exceptionally high βcat levels in all cells (*Figure 9D*). We scored mutant embryos blinded to genotype. Consistent with its rescue of viability and cell fate, APC2DD fully rescued βcat destruction (*Figure 9E* vs *Figure 9F,K,O*; 30/30 APC2DD embryos were scored as wild-type). In contrast, APC2AA had substantially reduced destruction complex function (*Figure 9O*; 14/29 embryos scored as mutant, consistent with 50% zygotic rescue). Maternal/zygotic *APC2 APC1* double mutants expressing APC2AA did not totally lose destruction complex function or the ability to respond to Wg signal. Stripes of cells with relatively reduced βcat levels were still present, but overall levels of βcat were substantially elevated and interestingly, levels were especially elevated in a subset of cells receiving Wg signal (*Figure 9G–I,L–N*). Together, these data suggest that these two putative phosphorylation sites in B are critical for APC2 function in vivo.

## Discussion

Mutations in APC disrupt embryonic development from its onset, and are the first step in most colon cancers. Although we know APC functions in the destruction complex to help target βcat for degradation, its mechanism of action remains unknown. Our study offers novel insights into APC's function in the destruction complex, providing the first glimpses of the internal structure of the APC: Axin complex, and leading to a novel dynamic and testable model of a regulated catalytic cycle of destruction complex function. Based on our data we propose an explicit mechanistic model for APC2 function inside the destruction complex (*Figure 10*). In step 1, APC2 stabilizes destruction complex assembly via its Arm rpts and SAMPs motifs. The assembled destruction complex then facilitates βcat phosphorylation (step 2) and protects it from dephosphorylation. GSK3, which phosphorylates βcat, Axin and other sites on APC, also phosphorylates APC2's R2 and B (step 3; CK1 may also be involved, as it is in the other phosphorylation events in the complex). We hypothesize this induces a conformational change in APC2 that releases the Arm rpts from association with Axin (step 4); since only a subset of APC2 molecules in the destruction complex would release the Arm rpts at any given time, the overall destruction complex would remain stable. We propose this conformational change allows transfer of phospho-βcat to the E3-ligase (step 5). Alternately, GSK3 may regulate the release of a complex of APC and phospho-βcat, allowing APC to shield phospho-βcat from dephosphorylation (*Su et al., 2008*) and guide it to the E3-ligase. The catalytic cycle of APC would then be reset by dephosphorylation (step 6). This provides a testable model for the mechanisms regulating a key signaling pathway. It also may provide insights into the assembly and dynamics of other large multiprotein complexes that assemble via dynamic multivalent interactions involving motifs within intrinsically disordered regions, leading to phase transitions (*Li et al., 2012a*; *Toretsky and Wright, 2014*).

### APC stabilizes the Axin complex and alters complex assembly, thus increasing destruction complex activity

While textbook diagrams often depict the destruction complex as a four protein 1:1:1:1 Axin: APC: GSK3:CK1 complex, many lines of data suggest the functional destruction complex is a large multimeric complex (e.g., *Fiedler et al., 2011*). One mechanism involved was already known: the DIX domain of Axin polymerizes in a head-to-tail fashion, in which beta-sheet 2 of one DIX domain interacts with beta-sheet 4 of the next monomer, thus forming filaments, and this polymerization is essential for destruction complex function (*Schwarz-Romond et al., 2007*; *Fiedler et al., 2011*).

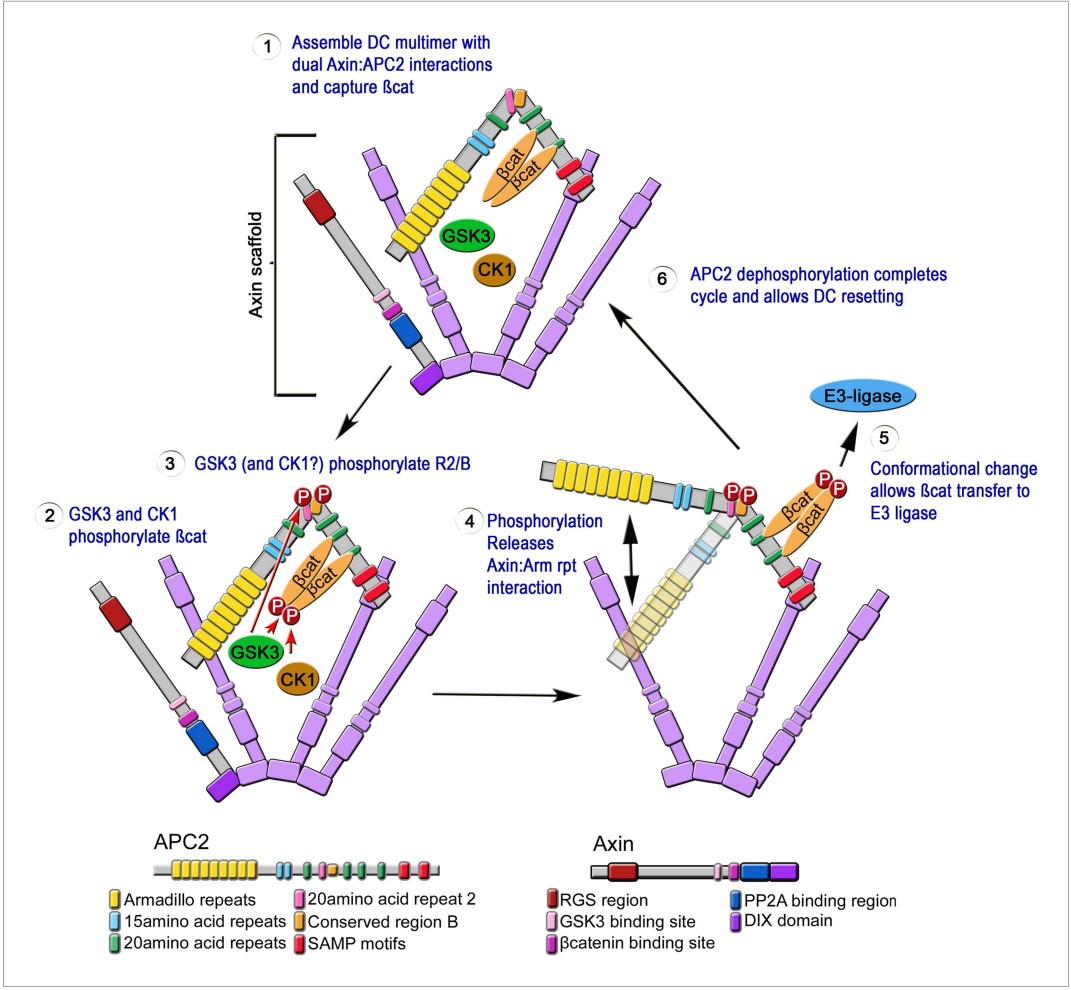

**Figure 10**. Speculative Model of APC2's catalytic cycle inside the destruction complex. (1) APC2 assembles with Axin via its Arm rpts and SAMPs. (2) APC2-bound βcat is phosphorylated by CK1 and GSK3. (3) GSK3 and CK1 phosphorylate R2/B region in APC2. (4) This induces a conformational change that releases APC2's Arm rpts from Axin. (5) APC2-bound βcat is released to the E3-ligase—alternately an APC–βcat complex is released. (6) Dephosphorylation resets the cycle.

These Axin polymers are responsible for assembly of the cytoplasmic puncta we use as a model. Previous and current data from a number of labs are consistent with the idea that the puncta serve as useful models of the smaller endogenous destruction complexes (*Faux et al., 2008*), based on correlations between puncta formation, dynamics, and function in βcat destruction. We used the puncta to visualize assembly and dynamics of Axin:APC complexes in parallel with functional studies in colon cancer cells and *Drosophila* to define APC's role in destruction complex assembly and function. It is important to note that our data in SW480 cells involve significant over-expression—it will be useful in the future to examine destruction complex structure and function at endogenous levels, perhaps by tagging endogenous loci using CRISPR.

APC is absolutely essential for the destruction complex to reduce βcat levels when Wnt signaling is off (*Mendoza-Topaz et al., 2011*), but the mechanism by which it acts remained unclear. Our FRAP and super-resolution microscopy data support a model in which one role of APC is to promote/stabilize Axin self-assembly and slow Axin turnover in the destruction complex, thus increasing destruction complex multimerization and its ability to process βcat. We found APC does so by interacting with Axin via two different kinds of interaction sites: the known direct interaction with the SAMPs, and a novel interaction via APC's Arm rpts, which may be direct or indirect. Both interactions

are critical for targeting βcat for destruction since APC without either SAMPs or Arm rpts cannot reduce βcat levels effectively. Almost all truncations in colon cancers remove the SAMPs (*Kohler et al., 2008*). Earlier data from our lab also implicated APC's Arm repeats in Wnt regulation—our new results provide a mechanistic basis for this effect.

APC2:Axin complexes were previously only resolved as co-localized spots. We provide the first glimpses inside the destruction complex. Super-resolution microscopy revealed that Axin puncta consist of Axin cables/sheets, which we hypothesize are bundled Axin polymers, assembled by the previously observed DIX domain polymerization (*Schwarz-Romond et al., 2007*). Consistent with our observation that APC2 promotes growth and reduces dynamics of Axin complexes, APC2 cables intertwine with and bridge Axin cables. Both Axin interaction sites are required for these effects. Stimulating growth of Axin complexes via destruction complex stabilization by APC2 may be essential when proteins are expressed at endogenous levels, increasing local concentrations of all destruction complex components, accounting for the highly efficient βcat destruction observed in the presence of APC. It will be interesting to explore the nature of the protein network involved at even higher resolution. As noted above, thus far we have visualized complexes of APC2 and Axin expressed at significantly elevated levels—it will be important to verify and extend these studies to the more modest size complexes found in vivo during normal development, using CRISPR to tag endogenous loci.

## Axin and APC cooperate to ensure efficient βcat destruction

APC mutations in tumors do not eliminate APC; instead the N-terminus, Arm rpts and some βcat binding sites remain (*Kohler et al., 2008*). Truncations cluster in the mutation cluster region (MCR), suggesting this region is critical. R2 and B, which are removed by truncations in the MCR, are essential for βcat downregulation (*Kohler et al., 2009*; *Roberts et al., 2011*). In the textbook model, the destruction complex phosphorylates βcat and thus targets it to an E3 ligase. It was thus surprising that colon cancer cells with truncations disrupting R2/B have high levels of phosphorylated βcat, in contrast to tumors retaining R2/B (*Yang et al., 2006*). Why do cells with non-functional APC have high levels of phosphorylated βcat?

Our data, together with earlier work, suggest Axin, the scaffold of the destruction complex, can mediate βcat phosphorylation even in cells with truncated APC, like the SW480 cells we use as a model, in which R2 and B are lost. Overexpressing Axin reduced total βcat levels, suggesting Axin can partially compensate for APC truncation, but phospho-βcat levels remained elevated inside the destruction complex. Interestingly, introducing APC2 reduced βcat accumulation in puncta and reduced phospho-βcat levels. These data suggest that without functional APC, Axin can mediate βcat phosphorylation, but transfer of βcat out of the destruction complex toward destruction is less efficient. It remains to be determined whether the accumulated phospho-βcat is actively transferred by Axin to the E3-ligase in the absence of functional APC, or if it is passively transferred due to a substantial increase in phospho-βcat. Thus while Axin can template βcat phosphorylation and can, at least in the presence of the truncated APC1 present in tumor cells, send it on to destruction, our data suggest APC promotes the rate at which βcat is transferred out of the destruction complex and sent to the proteasome. Future use of photoactivatible βcat constructs will further clarify this.

Based on our model APC mediated βcat transfer is only possible when R2 and B are maintained in the truncated APC (*Figure 10*). R2 and B are essential for function in the absence of endogenous APC function in vivo in *Drosophila* (*Roberts et al., 2011*). In SW480 cells our data suggest they are essential for APC2 to further simulate the rate of βcat destruction mediated by Axin transfection. Consistent with this, tumor cells that retain R2/B in the truncated APC have low levels of phospho-βcat (*Yang et al., 2006*), suggesting APC is still able to facilitate βcat transfer out of the destruction complex due to the presence of the Arm rpts, R2 and B (although the transfer would be less efficient due to loss of the SAMPs, the Axin binding sites). In contrast, truncated APC mutants that disrupt R2 and B function can associate with the Axin complex via the Arm rpts, but would not be able to assist in βcat transfer out of the destruction complex. The development of CRISPR knockout technology will allow future examination of the importance of truncated APC1, as well as the endogenous APC2 and Axin expressed in colon cancer cells in destruction complex assembly and function.

## Destruction complex function requires dynamic APC2:Axin interactions regulated by R2/B

APC's Arm rpts bind cytoskeletal regulators (*Nelson and Nathke, 2013*), but their mechanism of action in Wnt signaling remained unclear. The fact that overexpressing hAPC1 fragments lacking the Arm rpts in SW480 cells rescued Wnt regulation initially suggested the Arm rpts were dispensable (*Rubinfeld et al., 1997*). However these fragments were only tested in the presence of the truncated endogenous hAPC1 in these cells, which retains the Arm rpts. Thus hAPC1 fragments without Arm rpts may work with endogenous truncated APC, restoring partial function. In contrast, in flies, in the complete absence of endogenous APC, APC2 requires its Arm rpts for Wnt regulation (*McCartney et al., 2006*; *Roberts et al., 2012*). Our data provide the first mechanistic role for the Arm rpts in Wnt function, demonstrating they act as a regulated Axin interaction site, and revealing that this interaction is conserved in humans. Whether this interaction is direct or indirect remains to be determined.

What then is the mechanism by which APC facilitates βcat destruction? Two conserved APC motifs, R2 and B, are essential to target βcat for degradation (*Kohler et al., 2009*; *Roberts et al., 2011*). Deleting either leads to βcat accumulation in the destruction complex, suggesting R2 and B are critical for destruction complex throughput of βcat. Our colocalization, FRAP, and co-IP assays further suggest R2/B controls APC2:Axin association via APC2's Arm rpts. These data are consistent with a model in which APC2's R2 and B trigger an intramolecular conformational change in APC2, releasing the Arm rpts from association with the Axin complex (*Figure 10*). In contrast, the SAMPs bind Axin independently of R2/B. In our model binding via the SAMPs would keep APC associated with Axin complexes when the Arm rpts are released, maintaining a functional destruction complex and facilitating βcat transfer to the E3-ligase. However, as noted above, it is also possible GSK3 may regulate the release of a complex of APC and phospho-βcat, allowing APC to shield phospho-βcat from dephosphorylation (*Su et al., 2008*) on the way to the E3-ligase.

## GSK3 phosphorylation of R2 and B may be the trigger for releasing the APC Arm repeat:Axin interaction

Motifs R2 and B include highly conserved serine/threonines matching the GSK3 and CK1 consensuses, and this region is phosphorylated by GSK3 in vitro. GSK3 increases APC2 dynamics, destabilizing the Arm rpts:Axin association via a mechanism that requires R2 and B. Strikingly, mutating two conserved serines in B to alanine blocked APC2's ability to reduce βcat levels, while a parallel phosphomimetic mutant did not disrupt APC2 function. We saw a similar reduction in APC2 function after mutating a conserved serine residue in R2. Consistent with our data, CK1epsilon phosphorylation of hAPC1 R2 occurs in an Axin-dependent fashion, and site directed mutagenesis blocking phosphorylation of two conserved serines in R2 (hAPC1 S1389 and S1392; distinct from and just C-terminal to the residue we mutated in APC2 R2) reduced the ability of a human APC1 fragment to down regulate Wnt signaling (*Rubinfeld et al., 2001*), further suggesting R2 phosphorylation also is important for APC function. Our data are consistent with a model in which GSK3 phosphorylation of R2 and B could be one major regulatory step in APC2's dynamic cycle in the destruction complex, triggering release of the Arm rpts from the Axin complex, and thus allowing APC2 to promote βcat release for destruction (*Figure 10*). However, our data also show that mutating two residues in motif B to prevent their phosphorylation had only a subtle effect on APC2 dynamics. This may suggest additive roles for multiple phosphorylated residues, or may suggest the connection between phosphorylation, dynamics and function is more complex. Further, GSK3 can phosphorylate most of the other proteins in the destruction complex, and thus it clearly plays multiple roles in its function.

Phosphorylation of R2/B itself could trigger release of APC's Arm rpts from the Axin complex, or alternately, phosphorylation may create a binding site for a binding partner that carries out this function. Intriguingly, a recent study found that α-catenin co-IPs with motif B of hAPC1, and suggests α-catenin is an essential player in destruction complex function (*Choi et al., 2013*). Perhaps R2/B phosphorylation by GSK3 regulates association of α-catenin with APC, and its binding induces alterations in APC:Axin interactions. It will be exciting to test this hypothesis.

The multiple potential phosphorylation sites in R2/B, the hypomorphic nature of our double point mutant in *Drosophila* vs the complete loss of function after deleting either R2 or B (*Roberts et al., 2011*), and the residual function of our two-residue phosphomimetic mutant may suggest different combinations

of phosphorylated residues in R2 and B help tune and regulate APC function in the destruction complex. Our results also suggest GSK3 is likely to affect destruction complex structure and dynamics via several mechanisms, consistent with its known role in phosphorylating Axin and other sites in APC. Mathematical modeling suggests that a reduction in GSK3 ability to phosphorylate βcat is one key step in Wnt signaling activation, placing GSK3 activity in the center of 'destruction complex inactivation' (*Lee et al., 2003*). It is intriguing to speculate about GSK3-mediated APC regulation in the context of Wnt signaling. Since GSK3 activity is inhibited by Wnt signaling, key residues in R2 and B that drive APC:Axin interaction dynamics would no longer be phosphorylated. Our data suggest this would decrease APC dynamics in the Axin complex (since APC's Arm rpts would more strongly associate with Axin), perhaps reducing βcat transfer out of the destruction complex. Thus the drop in GSK3 activity upon Wnt signaling would not only downregulate the destruction complex via its main target, βcat, but through a core destruction complex component, APC, by acting via R2 and B. It will be of interest to explore further these complex interactions in both Wnt off and Wnt on states.

One intriguing property of the APC2AA mutant in *Drosophila* is that Wnt regulation of βcat was maintained, but βcat levels were elevated in <u>both</u> Wnt off and Wnt on regions, presumably reflecting a less efficient destruction complex. This re-focused our attention on the much higher levels of βcat seen in *APC2 APC1* mutants than occur in wildtype cells where Wg signal turns the destruction complex 'OFF'. These results suggest the destruction complex remains active when Wnt signal is on, but operates less effectively—this is very consistent with recent in vitro work (*Hernandez et al., 2012*). A destruction complex operating at reduced levels could lead to high enough βcat levels to activate Wnt target genes but could maintain low enough levels to be quickly shut down when needed; it would also prevent the apoptosis seen in some cell types at extremely high βcat levels. The drop in GSK3 activity upon Wnt signaling would both decrease phosphorylated βcat and, based on our model, inhibit APC facilitating βcat transfer to the E3-Ligase. This speculative hypothesis would merge two recent studies proposing either reduced βcat phosphorylation (*Hernandez et al., 2012*) or a key regulated step involving transfer of βcat to the E3-ligase (*Li et al., 2012b*) as key steps in Wnt regulation of the destruction complex. It will be intriguing to further probe the role of phosphorylation of R2/B in the mechanism of destruction complex action.

## Materials and methods

### Constructs

*Drosophila* APC2, human APC's Arm rpts (1–1012aa), and Axin were cloned as in *Roberts et al. (2011)*. In short, constructs were cloned into pECFP-N1 (Clontech, Mountain View CA) via Gateway (Invitrogen, Waltham MA) with either an N-terminal 3XFlag-tag or GFP-tag (*Roberts et al., 2011*). The C-terminal-RFP vector was generated by cloning the Gateway cassette and TagRFP from pTag-RFP (Evrogen, Russia) into pECFP-N1. For Phospho-APC2 constructs Serine 688 and 692 were changed using PCR stitching. *Drosophila* Axin fragments were N-term (1–54aa), RGS (55–171aa), GSK3 binding region (171–494aa), βcat binding region (494–532aa), PP2A binding region (532–666aa), and DIX (666–747aa).

### Antibodies

1° antibodies: βcat (BD Transduction, San Jose CA, 1:1000), FlagM2 (Sigma, St. Louis MO, 1:1000, 1μg/ml IP), GFP (Novus Biologicals, Littleton CO, 1:2000), βcat-S33/37 (Abcam, UK, 1:1000), γ-tubulin (Sigma, 1:5000), tagRFP (Evrogen, 1:5000), hAPC1 (Calbiochem, Billerica MA, 1:1000), hAxin1 (Cell Signaling, Danvers MA, 1:2000), aPKCγ (Santa Cruz Biotechnology, Dallas TX, 1:2000). 2° antibodies: Alexa 568 and 647 (Invitrogen, 1:1000), HRP anti-mouse and anti-rabbit (Pierce, Rockford IL, 1:50,000).

### Cell culture

SW480 cells were cultured in L15 medium (Corning, Tewksbury MA) + 10% heat-inactivated FBS+1X Pen/Strep (Gibco, Waltham MA) at 37 °C without $CO_2$. Lipofectamine 2000 (Invitrogen, Waltham MA) was used for transfections following manufacturer's protocol. For immunostaining and IP cells were processed after 24 hr. Immunostaining was as described in *Roberts et al. (2011)*. In short cells were fixed 5 min in 4% formaldehyde/1XPBS, rinsed 5 min in 0.1% Triton-100/1XPBS, blocked with 1% normal goat serum/1XPBS, and incubated in antibody. Samples were mounted in Aquapolymount (Polysciences, Warrington PA). For drug treatment 30 μM LiCl (Sigma; dissolved in L15), 2 μM BIO (Tocris, UK) or 25 μM MG132 (Calbiochem; both dissolved in 99% EtOH) were added 24 hr after transfection, and incubated 6h.

## Microscopy

Immunostained samples were imaged on a LSM Pascal microscope (Zeiss) and processed with the LSM image browser (Zeiss, Germany). SIM microscopy was carried out on the Deltavision OMX (GE Healthcare Life Sciences, Pittsburgh PA) using 4% formaldehyde fixed samples mounted in Vectashield (Vector, Burlingame CA) following manufacturer's protocol. Images were processed using Imaris 5.5, ImageJ and the LSM Image Browser. PhotoshopCS4 (Adobe, San Jose, CA) was used to adjust levels so that the range of signals spanned the entire output grayscale and to adjust brightness and contrast.

## Estimating relative over-expression levels of Axin and APC

We roughly calculated levels of expression by comparative immunoblotting, suggesting that expression levels were in the order: hAxin1-GFP > fly Axin-GFP > endogenous hAxin1. Since fly Axin is not recognized by hAxin1 antibodies we did this in two steps, first comparing levels of tagged human Axin1 vs *Drosophila* Axin using antibodies against the GFP epitope tag, and then comparing the levels of the transfected human Axin1 vs the endogenous Axin1 protein in SW480 cells, using antibodies against human Axin1 (*Figure 1—figure supplement 1C*). Because differences in expression levels took us out of the linear range of film, we diluted the more concentrated sample by a known amount. Protein bands in immunoblots of diluted samples (*Figure 1—figure supplement 1C3–4*) were quantitated in ImageJ. Levels were normalized by (1) determining γ-tubulin levels in *Figure 1—figure supplement 1C1–2* where samples were equally loaded, (2) by measuring γ-tubulin of diluted samples, and by calculating in the dilution factor. Once all samples were normalized to γ-tubulin the ratio between GFP-tagged fly and human Axin was calculated. Next, the ratio between hAxin1-GFP and endogenous hAxin1 was determined. Lastly the overexpression levels of fly GFP-Axin to endogenous hAxin1 were determined using the formula: ((ratio GFP-FlyAxin to hAxin1-GFP) × (ratio hAxin1-GFP to endogenous hAxin1)) / Transfection efficiency.

We then carried out a similar procedure for APC2, comparing levels of tagged human APC1 cloned so as to mimic the truncated APC1 seen in SW480 cells vs tagged *Drosophila* APC2 using antibodies to the Flag epitope, and then levels of tagged truncated human APC1 vs that of the endogenous truncated APC1 protein (*Figure 1—figure supplement 1D*). Lastly the overexpression levels of fly GFP-Axin to endogenous hAxin1 were determined using the formula ((ratio Flag-fly APC2 to Flag-hAPC1-1338) × (ratio Flag-hAPC1-1338 to endogenous hAPC1-1338))/transfection efficiency.

## Quantification

Z-projections of cell image stacks were generated using ImageJ. βcat fluorescent intensity: Cells were outlined, mean intensity measured, background subtracted, and βcat average intensity of a transfected cell normalized to mean of the βcat intensity of 2–3 adjacent untransfected cells. 10 cells were each measured in 3 independent experiments. Puncta colocalization of APC2ΔSAMPs with Axin in BIO/LiCl treated cells were determined by scoring for puncta formation in the APC2ΔSAMPs channel. 100 cells were scored in 3 independent experiments. APC:Axin complex size: Particle Analyzer of ImageJ was used. Background was subtracted and threshold for particles set to 200. Cytoplasmic puncta of 10 cells were averaged. Cell images were taken with LSM Pascal (Zeiss) with a resolution of 5.7 pixel/μM. Mean number of particles per cells was calculated from size measurements. Puncta volumes were measured using Imaris Software (Bitplane, Concord MA) from image z-stacks acquired on the Deltavision OMX (GE Healthcare Life Sciences). For comparing sequences, ClustalW2 (EMBL, Germany) was used for alignment. Statistical tests used the Student's *t*-test.

## FRAP

FRAP was conducted using Eclipse TE2000-E microscope (Nikon, Japan) 24–72 hr after transfection. Drug treated samples were measured 6–48 hr after drug treatment (30–72 hr after transfections). Movies were taken at 1 frame/3 s or 1 frame/6 s for 20 min and bleaching was conducted for 8 s with 100% laser. Movies were processed using FRAP analyzer in ImageJ. Bleached area and cell were outlined, background was subtracted and the movie was processed with FRAP profiler. Values were normalized and recovery plateau and standard error were calculated by averaging 10 movies. For $t_{1/2}$ values were processed in GraphPad (La Jolla CA) using non-linear regression (curve fit)-one phase decay. $t_{1/2}$ of 10 movies was averaged and standard error calculated.

## Protein work

IPs were as described in *Li et al. (2012b)*. In short cells were lysed on ice 15 min in 150 mM NaCl, 30 mM Tris pH 7.5, 1 mM EDTA, 1% Triton-X-100, 10% glycerol, 0.5 mM DTT, 0.1 mM PMSF + proteinase/phosphatase inhibitors (EDTA-free, Pierce), lysates cleared by centrifugation at 13,200 rpm 30 min at 4 ˚C, and preincubated with Sepharose beads 2h at 4 ˚C. 1 µg/ml antibody was added and incubated overnight at 4 ˚C. Beads were blocked in 5% BSA/1xPBS overnight at 4 ˚C, added to antibody-lysis mix, incubated 1–2 h at 4 ˚C, washed 5× with lysis buffer 4 ˚C, mixed with 2xSDS and incubated 10 min at 96 ˚C . Cell lysis for immunoblotting was similar. For βcat protein levels centrifugation was at 13,200 rpm for 30 min or 1000 rpm for 10 min at 4 ˚C. Drug treated cells were harvested after 6h. Proteins were run on 8% or 6% (hAPC1) SDS gels and blotted to nitrocellulose. IP quantification was in ImageJ using Gel plot analyzer. IP baits were normalized to amount of IPed protein, and βcat protein levels were normalized to loading control γ-tubulin.

## In vitro kinase assay

R2/B fragments of hAPC1 (1355aa–1465aa) and fly APC2 (632aa–733aa) were cloned via Gateway into pdest15-GST tagged vector. Protein expression was induced via IPTG (Apex, San Diego CA) and protein purification was conducted using Glutathione beads (Sigma). Kinase assays using human GSK3β (Sigma#G4296) were conducted following the manufacturer's protocol. The GSK3 substrate peptide YRRAAVPPSPSLSRHSSPHQ(pS)EDEE (based on human muscle glycogen, Signalchem, Canada) was used as a positive control. Samples were run using Tricine SDS-PAGE (*Schagger, 2006*), and the gel was Coomassie stained and measured for radioactivity by exposing to film.

## Fly work

Transgenic fly lines were generated by Best Gene (Chino Hills, CA). APC2 transgenes were crossed into the *APC2$^{g10}$ APC1$^{Q8}$* double mutant backgrounds as described previously (*McCartney et al., 2006*). Maternal/zygotic double mutants for both APCs were generated using the FRT/FLP/DFS technique (*Chou and Perrimon, 1996*). Fly crosses and heat-shock conditions were as described in *Roberts et al. (2011)*. Embryonic lethality and cuticle preparations were conducted as described in *Wieschaus and Nüsslein-Volhard (1998)*.

# Acknowledgements

Thanks to K. Bloom for advice on FRAP, M. Price for advice in 3D imaging, P. Pellett and T. Kreipe for help with the OMX superresolution microscopy, B. Major, B. Duronio, D. Roberts, V. Bautch, K. Slep, A. Fanning and the anonymous reviewers for helpful comments and R. Cheney and Alakananda Das for reagents and technical advice.

# Additional information

## Funding

| Funder | Grant reference | Author |
| --- | --- | --- |
| National Institute of General Medical Sciences (NIGMS) | R01 GM47857 | Mark Peifer |
| Howard Hughes Medical Institute (HHMI) | HHMI International Student Research Fellowship (no grant number) | Mira I Pronobis |
| National Heart, Lung, and Blood Institute (NHBLI) | 1ZIAHL006126 | Nasser M Rusan |

The funders had no role in study design, data collection and interpretation, or the decision to submit the work for publication.

## Author contributions

MIP, Conception and design, Acquisition of data, Analysis and interpretation of data, Drafting or revising the article; NMR, Acquisition of data, Analysis and interpretation of data, Drafting or revising the article; MP, Conception and design, Analysis and interpretation of data, Drafting or revising the article

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
