## [Decision Letter]

Thank you for submitting your work entitled “A novel GSK3-regulated APC:Axin interaction regulates Wnt signaling driving a catalytic cycle of βcatenin destruction” for peer review at *eLife*. Your submission has been favorably evaluated by Fiona Watt (senior editor), a Reviewing editor, and three reviewers.

The reviewers have discussed the reviews with one another and the Reviewing editor has drafted this decision to help you prepare a revised submission.

The reviewers felt that this manuscript provided important new information about APC protein function, and that the super-resolution and FRAP studies provided new insight into the destruction complex. The reviewers appreciated the use of both *Drosophila* and tissue culture models to probe the system, and felt the *Drosophila* rescue experiments were elegant. However, there were many places where the reviewers felt that there were additional experiments and important controls needed to support the conclusions of the manuscript. In particular, providing more information about the levels of overexpression in tissue culture cell lines would be helpful. More support is also needed for the conclusion that APC dynamics promotes beta-catenin flux through the DC (alternative approaches are suggested by the reviewers, such as the use of proteasome inhibitors). More explanation is needed for the selection of the sites focused on, and more mutational analysis should be considered in order to strengthen their conclusions. In addition, there were many places where more restricted (less-broad) conclusions were merited by the data presented in this manuscript (as detailed fully below, in the reviewers' comments). As an online journal *eLife* provides sufficient space to fully explain both rationales for and the important caveats of the experimental approaches used.

Detailed comments:

1) All of the cell biology analysis is done on punctae that are only observed when Axin or Axin/APC are overexpressed. There are a couple of general sentences stating that the field is comfortable with using these punctae as surrogates for the endogenous DC that are not very compelling. The super-resolution microscopy is cool and the images are gorgeous, though the fact remains they are large protein complexes caused by overexpression. They are probably are functional in regard to down-regulating beta-catenin, but are all their characteristics relevant to endogenous DC function? This is an inherent limitation of the manuscript. It would be helpful for readers to know what level of overexpression is being employed (compare to endogenous proteins) and what is variability of expression among individual cells transfected with APC or Axin constructs.

2) A key part of the authors' model is that APC dynamics promotes beta-catenin throughput through the DC. But the data supporting this in Figure 3 are unconvincing. The results in Figure 3 that APC2 enhances Axin's ability to downregulate beta-catenin isn't informative: APC2 alone is as good as APC2 + Axin. This undercuts the utility of the Figure 3 comparison, maybe there is undetectable levels of beta-catenin in Axin/APC2 punctae because there is less overall beta-catenin. The data with phospho-beta-catenin has the same fatal flaw. Perhaps using proteosome inhibitors would allow the authors to make more compelling conclusions, if conditions could be found where overexpression of APC and Axin did not affect beta-catenin levels.

3) The data in Figures 4, 5, 6 and 7 lead the authors to conclude that GSK3 acts through the R2 and B motifs of APC to regulate its association with Axin and beta-catenin turnover in the DC. They then proceed to the next critical test: mutating predicted GSK3 sites in APC and testing them functionally. They mutate two serines in B, converting them to either alanines (APC2AA) or aspartic acid (APC2DD; hoping for a phosphomimic effect). Clearly the AA mutant is defective in APC function, nicely shown in the fly rescues in Figure 9. But the effects on punctae dynamics are absent (Figure 8; comparing wild-type APC2 with APC2AA). The authors focus on the small difference between APC2DD and APC2AA, but this difference is also very, very subtle. The data don't fit into a coherent model, since they demonstrate that a surgical mutation that has a big effect on APC function has no detectable effect on APC/Axin dynamics. Perhaps a more comprehensive mutagenesis of the predicted phosphorylation sites would provide a clearer link between GSK3 phosphorylation of APC, APC dynamics in DC punctae and APC function. But the two mutants characterized are not sufficient.

4) The language of the Abstract and Discussion, proposing a GSK3 induced conformational change in APC is a bit overblown. It's very reasonable to invoke a phosphorylation induced conformational change: it's a common mechanism for regulating protein shape. But the manuscript doesn't provide direct evidence for a conformational change, and I don't think the data provides a paradigm for the broader field of researchers studying multiprotein complexes. Rather, I think this report does address important issues in APC function in the DC, and that is a significant goal.

5) Figure 1 shows that the *Drosophila* APC2 arm-repeat domain functions as a second means of interacting with Axin using co-IP and co-localization analysis. Since no direct binding interactions are shown, it is more accurate/clearer to state that “APC has two means of interacting with the Axin complex” rather than “with Axin”.

6) Figure 2 shows that APC2 and Axin form structured macromolecular complexes in vivo by SIM. APC2 nearly doubles the size/volume of the DC and there is a trade-off between DC punctae abundance and size. Since GSK3 inhibition leads to APC2 stabilization in the DC by virtue of activating a second Axin binding interaction (via Arm repeats)– does GSK3 inhibition reduce the abundance/increase the size of DC ‘factories’ in cells? I am trying to get a sense if/how size correlates with DC activity.

7) Figure 5 shows that APC2 R2 and B domains regulate APC2 dynamics and β-cat removal from the complex. Given the model in Figure 10, is the data here also compatible with a model where APC and β-cat are together released for the ubiquitylation step? This seems plausible especially give the sense that there is a seemingly abundant diffuse cytosolic pool of APC and β-cat relative to Axin? Such a model might also be able to better integrate previous studies showing that APC shields the N-terminus of β-cat from dephosphorylation (49).

8) Figure 5: Is the increase in β-cat retention at punctae due to increased loading on APC or Axin? Simple co-IP analysis might allow one to infer this-so long as most of the complex is soluble in gentle detergents.

9) Evidence for differences in β-cat “flux” through the complex is largely inferred from its steady state accumulation of β-cat within punctae, rather than any direct measurement of this by FRAP or perhaps better, photoconversion at the punta. Have the authors tried this? I ask only because the β-cat accumulation at punctae is somewhat confusing-one can only see this structure if the diffuse pool was effectively degraded, and all of the mutants basically need to do this in order for the authors to capture differences in β-cat accumulation within punctae. I think it would be helpful to clarify that all of the mutants can degrade β-catenin (as evidenced by depletion of the diffuse cytosolic pool of β-cat), but that there must be rate differences due to accumulation differences in the punctae.

10) It would be helpful if Figure 1 showed that domain B is also called the Catenin Interaction Domain (CID, [6]), and clearly define this since CID was not adequately described in the Introduction. Also, since all reviews cite that Axin binds CK1 in addition to GSK3, it would be helpful to have this interaction mapped for completeness (if precisely known)?

11) In the fourth paragraph of the Introduction it sounds like the Arm domain of APC behaves as an intrinsically disordered domain, but it seems more appropriate to use this description for the 15 and 20R downstream (especially since many β-cat binding ligands are ID in the region of binding. Please clarify or specify the regions in Figure 1 as “intrinsically disordered region, or IRD.”

12) To study the dynamics of APC-Axin interactions, the authors transfect various fly and human constructs into the well-known SW480 colon cancer cell line, which expresses a wild type copy of APC and a truncation of APC containing the first 1338 AA. In their analyses of Axin-APC interactions, they do not take into consideration the effect of endogenous proteins in mediating the interaction. The role of endogenous proteins is only briefly mentioned in the subsection “APC2 stabilizes the Axin complex and promotes DC throughput of βcat”.

13) In Figure 8 the authors identified putative GSK3 phosphorylation sites. It is unclear how these meet the consensus for GSK3 other than being Ser/Thr residues. Usually there is a priming site 4-5 amino acids C terminal to the putative target site. The authors need to define what criteria they used to identify putative sites. I have a number of concerns about this experiment: The kinase assay is lacking positive controls. What is the band in the GST/GSK3 lane? Why are there multiple bands in the R2/B/GSK3 lane? What do the asterisks mean in the figure? Why are there GST-only bands in the coomassie lanes 3-6? What was the rationale for mutating the two selected Ser residues? They don't seem to match the consensus any better than the others. The logic given states that they are the most N terminal, when in fact two other Ser residues are more N terminal, and there are no potential CK1 priming sites. As the authors wisely state in subsection “Mutating putative phosphorylation sites in B disrupts APC2's function in regulating βcat destruction in SW480 cells”, it is very likely that GSK3 inhibition affects many aspects of DC dynamics, so all findings should be interpreted cautiously. The *Drosophila* work with the phospho-resistant and phospho-mimetic transgenes is very elegant.

14) In their model, as described in the Discussion, the authors suggest that GSK3 and CK1 phosphorylate R2 and B motifs. They cannot make this conclusion, since B has no CK1 motifs and no kinase assays with CK1 were performed. They also state that Axin assembles in a head to tail fashion via its DIX domain. I would argue this is a tail to tail interaction.

---

## [Author Response]

The reviewers felt that this manuscript provided important new information about APC protein function, and that the super-resolution and FRAP studies provided new insight into the destruction complex. The reviewers appreciated the use of both Drosophila and tissue culture models to probe the system, and felt the Drosophila rescue experiments were elegant. However, there were many places where the reviewers felt that there were additional experiments and important controls needed to support the conclusions of the manuscript. In particular, providing more information about the levels of overexpression in tissue culture cell lines would be helpful. More support is also needed for the conclusion that APC dynamics promotes beta-catenin flux through the DC (alternative approaches are suggested by the reviewers, such as the use of proteasome inhibitors). More explanation is needed for the selection of the sites focused on, and more mutational analysis should be considered in order to strengthen their conclusions.

As we describe in detail below, we carried out additional experiments to address each of these three concerns.

*In addition, there were many places where more restricted (less-broad) conclusions were merited by the data presented in this manuscript (as detailed fully below, in the reviewers' comments). As an online journal* eLife *provides sufficient space to fully explain both rationales for and the important caveats of the experimental approaches used.*

These are all good points and we appreciated the opportunity to both expand explanations and soften some of our conclusions.

Detailed comments:

1) All of the cell biology analysis is done on punctae that are only observed when Axin or Axin/APC are overexpressed. There are a couple of general sentences stating that the field is comfortable with using these punctae as surrogates for the endogenous DC that are not very compelling. The super-resolution microscopy is cool and the images are gorgeous, though the fact remains they are large protein complexes caused by overexpression. They are probably are functional in regard to down-regulating beta-catenin, but are all their characteristics relevant to endogenous DC function? This is an inherent limitation of the manuscript. It would be helpful for readers to know what level of overexpression is being employed (compare to endogenous proteins) and what is variability of expression among individual cells transfected with APC or Axin constructs.

The reviewers and editors were correct that this is in important point that needed to be further clarified, and which needed to be pointed out as an existing caveat to these (and many previous) studies. We have examined both issues. As people working with cell culture transfection can appreciate (and as the Reviewer pointed out), there are a number of variables in addressing this issue, which together mean we can only give a range, not a hard and fast number – these include variability in transfection efficiency between different constructs, between different experiments, and variations in DNA uptake and thus protein expression between different cells in a single experiment. We thus took two different approaches to assess this.

To explore the effect of variability of expression among individual cells transfected with GFP or RFP-tagged APC or Axin constructs in a given experiment, individual cells were quantitated for the levels of GFP-APC2, Axin RFP or both (for Axin plus APC transfected cells), using fluorescence quantitation, and simultaneously were assessed for total ßcat levels, normalized to untransfected cells on the same slide (new Figure 1—figure supplement 1). This allowed us to determine if relative level of over-expression was important for the effect on ßcat levels. Strikingly, Axin was less efficient at modulating ßcat levels than both “APC alone” and than APC+Axin at all levels of expression – thus differences between the function of Axin versus APC2+Axin are not seen only at the highest levels of expression. Second, at the higher levels of expression, the constructs actually became less effective at mediating ßcat destruction. This may be consistent with the idea suggested previously by the Kirschner lab that at very high levels of expression, over-expressed proteins might form partial, non-functional complexes. It also may reflect the cytoplasmic retention of ßcat we previously observed by APC2 constructs that retain ßcat binding sites (41).

Second, we measured rough absolute levels of over-expression. Since we were transfecting fly proteins into human cells, we could not simply use human antibodies to answer this question. We thus used transfected human APC1 (truncated to be of the size of the endogenous truncated APC1 found in SW480 cells) and transfected human Axin1 as intermediates. Transfected Flag-tagged fly APC2 and Flag-tagged human APC1 or transfected GFP-tagged fly Axin or human Axin1 could thus be detected with the same anti-Flag or GFP antibodies and their levels compared by immunoblotting.

Transfected human APC1 could then be compared to endogenous truncated human APC1, or transfected human Axin1 compared to endogenous human Axin 1 using antibodies to the endogenous proteins. In each case, these numbers were then adjusted for transfection efficiency. Three independent experiments were assessed for each – representative Western blots are now presented in new Figure 1—figure supplement 1, and the full set of quantitations is in new Table 1. The results are reported in the first section of the Results, and the procedure used is described in detail in the Figure legend and the Methods.

For Axin, we could compare transfected *Drosophila* Axin to endogenous wild-type human Axin1 found in SW480 cells. Here the level of overexpression was 80-120 fold. This issue was more complex in the case of APC, as SW480 cells do not accumulate wild-type APC1 – instead they accumulate a truncated APC1 ending at amino acid 1338. We thus compared transfected fly APC2 to this truncated human APC1 protein, seeing ratios of roughly 1400-2400 fold. However, we think this is likely to significantly overestimate over-expression relative to what would be the normal level of accumulation of wild-type human APC1 in a normal colon cell for two reasons. First, the truncated APC1 protein present in SW480 cells is reasonably likely to accumulate at lower levels than wild-type APC1, as it is likely to be subjected to nonsense-mediated mRNA decay like many other proteins with early stop codons. In fact, a recent paper noted that 12/14 APC- mutation–positive families in whom the mutation was located before the last exon of the gene had allele specific imbalance at the mRNA level, which is consistent with a mechanism of nonsense-mediated decay (5). Second, we also may need to reduce the difference by a factor of two as it is probable SW480 cells carry only one copy of the truncated APC allele, as most colorectal tumors either have the second allele mutated early enough to not produce a truncated protein or have lost the second allele by deletion (8).

We now include these new data at the beginning of the Results section, outlining things as described above, and ending by noting this is an important caveat of the study as follows: “Regardless, it is important to remember that while the puncta provide a useful and visualizable model of the destruction complex, we are examining over-expressed proteins.” We added a similar caveat in the relevant section of the Discussion, as follows: “It is important to note that our data in SW480 cells involve significant over-expression – it will be useful in the future to examine destruction complex structure and function at endogenous levels, perhaps by tagging endogenous loci using CRISPR.”

In considering these issues, it is useful to remember that in our previous work (and that of many other labs) there are many constructs that we and others have tested that cannot downregulate ßcat, even when similarly over-expressed, including the constructs tested here with single or double point mutants in serine residues. It is also useful to note that with one exception (APC2∆SAMPS), in our previous work (41) and extensions of this (53) there has been a perfect correspondence between function of constructs in restoring targeting ßcat for destruction when overexpressed in SW480 cells and when expressed at endogenous levels in *Drosophila*.

Finally, after submitting the manuscript we became aware of a relatively recent paper from the Bienz lab that further supports the idea that puncta formation is a normal feature of Axin, and that one need not substantially over-express it to be able to easily visualize puncta. [10] explored the effects of tankyrase inhibitors on SW480 cells. They found that tankyrase inhibitors increased levels of AXIN13-5x and AXIN2 5-20x and this was sufficient to trigger formation of Axin-puncta (interestingly, ßcat also accumulated in these puncta to detectable levels). We now note this at the beginning of the Results section that describes the puncta.

*2) A key part of the authors' model is that APC dynamics promotes beta-catenin throughput through the DC. But the data supporting this in*
Figure 3
*are unconvincing. The results in*
Figure 3
*that APC2 enhances Axin's ability to downregulate beta-catenin isn't informative: APC2 alone is as good as APC2 + Axin. This undercuts the utility of the*
Figure 3
*comparison, maybe there is undetectable levels of beta-catenin in Axin/APC2 punctae because there is less overall beta-catenin. The data with phospho-beta-catenin has the same fatal flaw. Perhaps using proteosome inhibitors would allow the authors to make more compelling conclusions, if conditions could be found where overexpression of APC and Axin did not affect beta-catenin levels.*

We have addressed this in two ways. First, as the reviewer notes, our data showed that expression of either “APC2 alone” or APC2 plus Axin similarly stimulated ßcat exit out of the puncta to destruction. In our initial manuscript, we had suggested this was “presumably due to the presence of endogenous human Axin”. We now have tested whether fly APC2 can and does interact with human Axin. We found that Fly GFP-APC2 colocalizes in puncta with exogenous hAxin1-RFP (present in new Figure 3), and more importantly, endogenous hAxin1 coIPs with Flag-APC2 expressed in Sw480 cells (new Figure 3). Thus this is at least a reasonable possibility.

Second, we followed the reviewers’ suggestion and used proteasome inhibitors to slow ßcat destruction in cells expressing APC2 plus Axin. As we show in new Figure 3—figure supplement 1, proteasome inhibition allows ßcat to detectably accumulate in the puncta of cells expressing APC2 + Axin (Figure 3—figure supplement 1). However, while proteasome inhibition elevates ßcat levels both in cells expressing Axin alone and in those expressing APC2 + Axin (Figure 3—figure supplement 1), cells expressing Axin still accumulate significantly higher levels of ßcat than those expressing Axin + APC2 (Figure 3—figure supplement 1). This suggests the difference in puncta accumulation is not solely due to the elevated ßcat levels. We also have softened the conclusion from this experiment, suggesting it is “consistent with our model”. The new text now reads: “One potential caveat to the increased accumulation of βcat in puncta of cells expressing Axin alone vs those expressing APC2 plus Axin is that the former cells may simply have higher overall levels of βcat, thus resulting in higher accumulation in puncta. To address this, we inhibited βcat destruction using the proteasome inhibitor MG132. As others have previously observed (44), proteasome inhibition elevates ßcat levels. Proteasome inhibition elevates ßcat levels both in cells expressing Axin alone and in those expressing APC2 plus Axin (Figure 3—figure supplement 1).

Strikingly, this allows it to accumulate in puncta even in cells expressing Axin + APC2 (Figure 3—figure supplement 1). However, cells expressing Axin still accumulate significantly higher levels of ßcat than those expressing Axin plus APC2 (Figure 3—figure supplement 1). Together, these data are consistent with a model in which APC increases βcat throughput of the destruction complex by stabilizing Axin assembly.”

*3) The data in*
Figures 4, 5, 6 and 7
*lead the authors to conclude that GSK3 acts through the R2 and B motifs of APC to regulate its association with Axin and beta-catenin turnover in the DC. They then proceed to the next critical test: mutating predicted GSK3 sites in APC and testing them functionally. They mutate two serines in B, converting them to either alanines (APC2AA) or aspartic acid (APC2DD; hoping for a phosphomimic effect). Clearly the AA mutant is defective in APC function, nicely shown in the fly rescues in*
Figure 9*. But the effects on punctae dynamics are absent (*Figure 8*; comparing wild-type APC2 with APC2AA). The authors focus on the small difference between APC2DD and APC2AA, but this difference is also very, very subtle. The data don't fit into a coherent model, since they demonstrate that a surgical mutation that has a big effect on APC function has no detectable effect on APC/Axin dynamics. Perhaps a more comprehensive mutagenesis of the predicted phosphorylation sites would provide a clearer link between GSK3 phosphorylation of APC, APC dynamics in DC punctae and APC function. But the two mutants characterized are not sufficient.*

These are important points and we have done a number of different things to address them. Before we describe them in detail, we would like to address the larger issue of what we have and have not learned from the work presented here and how we have clarified these issues in our revised manuscript.

New insights that precede this section of the manuscript: It is clear from earlier work from our and other labs that R2 and B are essential for APC function in ßcat destruction in both cultured cells and in vivo in *Drosophila*. In Figures 2 and 3 we show that APC alters the structure of macromolecular complexes formed by Axin (as assessed by SIM) and alters Axin mobility in Axin puncta. In Figure 4 we show that these effects require both the Arm rpts and the SAMPs. These data provide novel insights into roles of APC in the destruction complex. In Figures 5 and 6 we address how R2/B fits into this picture, showing its normal role is to destabilize APC:Axin association, and that it does so by antagonizing association mediated by APC's Arm rpts. This provides novel insights into the role of R2/B in APC function. In Figure 7 we show that GSK3 inhibition stabilizes APC:Axin association; our data suggest GSK3 activity antagonizes association mediated by APC's Arm rpts, and thus GSK3 activity and R2/B appear to act either together or in parallel to regulate APC function. This provides a novel role for GSK3 in destruction complex function, adding to its earlier described roles.

Making the connection between the role of GSK3 and particular phosphorylation sites: In our initial manuscript, we then went on to explore the role of particular phosphorylation sites in this region. We chose a pair of serine residues in region B, and made either phosphomimetic or non-phosphorylatable variants. We then went on to test these functionally in both SW480 cells and in flies, showing, as the reviewer(s) points out, that the APC2AA is significantly reduced in function in both contexts.

As the reviewers correctly point out, the double point mutant does not fully mimic the effects on APC dynamics of either R2 or B deletion or of GSK3 inhibition. They point out that this suggests our model is over simplified. We agree that in our initial manuscript we over-emphasized the match between function and dynamics, and thus under-stated the complexities. We have addressed this issue in several ways:

First, we have matched the number of samples analyzed for dynamics of wildtype, phosphomimetic and nonphosphorylatable. This revealed that the recovery fraction of the non-phosphorylatable is significantly different from that of wild-type APC and also significantly different from that of the phosphomimetic mutant. The differences in dynamics between wild-type and the phosphomimetic mutant are not statistically significant. We also have toned down the text to be very careful we do not overstate the case here, making it clear both that there are likely other phosphorylation sites and that GSK3 likely has effects not mediated solely by phosphorylating R2/B (see below).

A) As suggested by the reviewers, we analyzed the effect on APC2 function in SW480 cells of a single point mutant in a putative phosphorylation site in R2. Strikingly, this point mutant also largely abolishes function in ßcat destruction. We also note in the Discussion that the Polakis lab had already analyzed a different double point mutant in R2 serine residues, and similarly saw diminished function in ability to downregulate ßcat levels. We agree that it would be good to analyze additional point mutants in *Drosophila* in the future, but this will be a time-consuming process that we feel extends beyond the scope of the current manuscript.

B) We significantly revised this section of the Results to more clearly state the caveats. Putting these changes together, this section of the Results now reads:

“R2 and B are essential for APC2's function in βcat degradation. Our hypothesis suggests phosphorylation of R2/B promotes release of APC2's Arm rpts from the Axin complex, and that this would be essential for the catalytic cycle of the destruction complex – thus mutating these putative GSK3 phosphorylation sites in motif B would reduce APC2 function in helping mediate ßcat destruction. To begin to test this, we replaced 2 (APC2AA; Figure 8, red arrows), 4 or 6 (Figure 8, magenta arrowheads) conserved serines in B that match the GSK3 consensus with alanine, to prevent phosphorylation. All reduced function in downregulating ßcat levels (see below and data not shown). We thus focused on the least altered of these, the mutant that replaced the more N-terminal two serine residues with alanine (APC2AA; Figure 8, red arrows), thus preventing phosphorylation. We also created a mutant that replaced these same two residues with aspartic acid, creating a phosphomimetic APC2 (APC2DD). Strikingly, while APC2DD effectively reduced βcat levels (Figure 8), APC2AA was unable to do so.

APC2AA cells accumulated βcat at levels as high or higher than adjacent untransfected cells (Figure 8), suggesting these two amino acid missense mutations substantially reduced APC2 function. APC2DD substantially reduced βcat levels (to ∼30%), but was not quite as effective as wildtype APC2 (Figure 8). This statistically significant difference may suggest dephosphorylation of these residues is also required for full APC2 function. We also found mutating a single conserved serine residue in R2 to alanine (APC2660; Figure 8) strongly diminished APC2 function in reducing ßcat levels (Figure 8). Together, these data are consistent with a model in which phosphorylation of conserved serines in APC2 motifs R2 and B are important for APC's function in the destruction complex to target βcat for degradation. Finally, we tested whether the first of these putative phosphomutants affected APC2 dynamics in the FRAP assay. Since GSK3 inhibition slowed APC's dynamics, we predicted APC2AA should have a lower turnover rate than wildtype APC or APC2DD. We saw a subtle but statistically significant reduction in APC2AA recovery fraction (Figure 8). However this was not nearly as dramatic as that of deleting R2 or B (Figure 5); perhaps this due to the fact that we only altered two of several potential phosphorylation sites. Together these data are consistent with the idea that phosphorylation of conserved serine residues in R2/B regulates APC's function in the destruction complex, but since the effect on dynamics was substantially less dramatic than that of deleting R2 or B, it suggests other residues in R2 and B may also contribute to regulating APC2 dynamics. Further, it is clear that GSK3 has other effects on the complex, complicating interpretation of its inhibition. It will be important to examine this in the future.”

We also softened this conclusion substantially in the Discussion. For example, one of the relevant sections now reads: “Motifs R2 and B include highly conserved serine/threonines matching the GSK3 and CK1 consensuses, and this region is phosphorylated by GSK3 in vitro. GSK3 increases APC2 dynamics, destabilizing the Arm rpts:Axin association via a mechanism that requires R2 and B. Strikingly, mutating two conserved serines in B to alanine blocked APC2's ability to reduce βcat levels, while a parallel phosphomimetic mutant did not disrupt APC2 function. We saw a similar reduction in APC2 function after mutating a conserved serine residue in R2. Consistent with our data, CK1epsilon phosphorylation of hAPC1 R2 occurs in an Axin-dependent fashion, and site directed mutagenesis blocking phosphorylation of two conserved serines in R2 (hAPC1 S1389 and S1392; distinct from and just C-terminal to the residue we mutated in APC2 R2) reduced the ability of a human APC1 fragment to down regulate Wnt signaling (43), further suggesting R2 phosphorylation also is important for APC function. Our data are consistent with a model in which GSK3 phosphorylation of R2 and B could be one major regulatory step in APC2's dynamic cycle in the destruction complex, triggering release of the Arm rpts from the Axin complex, and thus allowing APC2 to promote βcat release for destruction (Figure 10). However, our data also show that mutating two residues in motif B to prevent their phosphorylation had only a subtle effect on APC2 dynamics. This may suggest additive roles for multiple phosphorylated residues, or may suggest the connection between phosphorylation, dynamics and function is more complex. Further, GSK3 can phosphorylate most of the other proteins in the destruction complex, and thus it clearly plays multiple roles in its function.”

We also note: “The multiple potential phosphorylation sites in R2/B, the hypomorphic nature of our double point mutant in *Drosophila* vs the complete loss of function after deleting either R2 or B (41), and the residual function of our two residue phosphomimetic mutant may suggest different combinations of phosphorylated residues in R2 and B help tune and regulate APC function in the destruction complex.

Our results also suggest GSK3 is likely to affect destruction complex structure and dynamics via several mechanisms, consistent with its known role in phosphorylating Axin and other sites in APC.”

4) The language of the Abstract and Discussion, proposing a GSK3 induced conformational change in APC is a bit overblown. It's very reasonable to invoke a phosphorylation induced conformational change: it's a common mechanism for regulating protein shape. But the manuscript doesn't provide direct evidence for a conformational change, and I don't think the data provides a paradigm for the broader field of researchers studying multiprotein complexes. Rather, I think this report does address important issues in APC function in the DC, and that is a significant goal.

We clearly became over-excited about this speculative possibility and thus carried away in our language. We removed the description of “conformational change” from the Abstract, revising it as follows:

“Our data suggest APC plays two roles: (1) APC promotes efficient Axin multimerization through one known and one novel APC:Axin interaction site, and (2) GSK3 acts through APC motifs R2 and B to regulate APC:Axin interactions, promoting high-throughput of βcatenin to destruction.”

We also have taken a look at the rest of the text and tried to restrict our use of the term conformation change to our description of the speculative model in the Discussion.

We also removed the overly broad claim about a paradigm from the Abstract – it now reads: “We propose a new dynamic model of how the destruction complex regulates Wnt signaling and how this goes wrong in cancer, providing insights into how this multiprotein signaling complex is assembled and functions via multivalent interactions.”

*5)*
Figure 1
*shows that the Drosophila APC2 arm-repeat domain functions as a second means of interacting with Axin using co-IP and co-localization analysis. Since no direct binding interactions are shown, it is more accurate/clearer to state that “APC has two means of interacting with the Axin complex” rather than “with Axin”.*

This was a good point. We changed the title of this Figure to: “APC2's Arm rpts provide a second means of interacting with the Axin complex.” and also have gone through the rest of the text to make it clear we do not know if this is a direct or indirect interaction, using the terms “association” or “interaction with the Axin complex”.

*6)*
Figure 2
*shows that APC2 and Axin form structured macromolecular complexes in vivo by SIM. APC2 nearly doubles the size/volume of the DC and there is a trade-off between DC punctae abundance and size. Since GSK3 inhibition leads to APC2 stabilization in the DC by virtue of activating a second Axin binding interaction (via Arm repeats)– does GSK3 inhibition reduce the abundance/increase the size of DC 'factories' in cells? I am trying to get a sense if/how size correlates with DC activity.*

This was an interesting issue, and we explored it as suggested. We inhibited GSK3 and measured both the size and number of APC2+Axin puncta. GSK3 inhibition did not further increase size or decrease number over the increase in size we saw by adding APC alone. This fits with our earlier analysis of deleting R2 or B – while each alteration stabilizes the APC-Axin interaction, it does not further increase puncta size or decrease puncta number. We also explored whether proteasome inhibition further increased the size or reduced the number of puncta. All of this new data in included in Figure 7 and in the Results as follows: “If GSK3 inhibition stabilizes the interaction of APC2 and Axin, and as noted above, APC2 also can stabilize Axin in puncta, increasing their size (and thus decreasing their number), then GSK3 inhibition might synergize with APC, further increasing the size of Axin puncta. We thus inhibited GSK3 with either BIO or LiCl in cells co-expressing Axin and APC2 and examined both puncta size and number. GSK3 inhibition did not further increase puncta size or further decrease puncta number (Figure 7). These data are consistent with our analysis above of APC2∆R2 and APC∆B – these mutations also stabilize APC2 in the destruction complex (Figure 5) but do not further increase puncta size or decrease puncta number (Figure 5—figure supplement 1). We also tested whether proteasome inhibition might trigger enlargement of the APC+Axin complexes – this also did not further increase puncta size or further decrease puncta number (Figure 7). Thus, while APC2 can stabilize Axin complexes, inhibition of GSK3 or the proteasome does not synergize with this.”

*7)*
Figure 5
*shows that APC2 R2 and B domains regulate APC2 dynamics and β-cat removal from the complex. Given the model in*
Figure 10*, is the data here also compatible with a model where APC and β-cat are together released for the ubiquitylation step? This seems plausible especially give the sense that there is a seemingly abundant diffuse cytosolic pool of APC and β-cat relative to Axin? Such a model might also be able to better integrate previous studies showing that APC shields the N-terminus of β-cat from dephosphorylation (*[49]*).*

This is an excellent suggestion – we agree it is a very plausible possibility and have now incorporated it into the description of our model in the Discussion as follows:

“Based on our data we propose an explicit mechanistic model for APC2 function inside the destruction complex (Figure 10). In step 1, APC2 stabilizes destruction complex assembly via its Arm rpts and SAMPs motifs. […] We propose this conformational change allows transfer of phospho-βcat to the E3-ligase (step 5). Alternately, GSK3 may regulate the release of a complex of APC and phospho-βcat, allowing APC to shield phospho-βcat from dephosphorylation (49) and guide it to the E3-ligase. The catalytic cycle of APC would then be reset by dephosphorylation (step 6). This provides a testable model for the mechanisms regulating a key signaling pathway.”

We made similar alterations later in the Discussion and in the legend for Figure 10.

*8)*
Figure 5*: Is the increase in β-cat retention at punctae due to increased loading on APC or Axin? Simple co-IP analysis might allow one to infer this-so long as most of the complex is soluble in gentle detergents.*

This is an interesting question. We attempted the experiment the Reviewer suggested, IPing tagged Axin co-expressed with wild-type APC2, APC2∆R2 or APC2∆B, and looking for ßcat co-IP. There was a trend toward increased ßcat in the IPs but due to the variability between IPs (likely due to, as suggested, to variability in solubility and in preservation of the complex upon lysis), we cannot make an air-tight statement about this. These data are now included as Figure 11. We would be happy to add them to the manuscript if the Reviewers and Editor think it appropriate.

Author response image 1.lnvestigating whether deleting R2 or B alters colP of þcat with Axin in SW480 cells. (A) Expression of Flag-Axin with indicated GFP-tagged APC2 constructs followed by lP of Flag-Axin and examination of the level of ßcat that colPs. Representative immunoblot of 3 independent experiments. (B) Quantification of 3 independent co-lPs. þcat levels were normalized to Flag-Axin. Levels of colPed ßcat were higher in 2 of 3 experiments with APC26R2 and in all three experiments with APC26B.**DOI:**
http://dx.doi.org/10.7554/eLife.08022.028

*9) Evidence for differences in β-cat “flux” through the complex is largely inferred from its steady state accumulation of β-cat within punctae, rather than any direct measurement of this by FRAP or perhaps better, photoconversion at the punta. Have the authors tried this? I ask only because the β-cat accumulation at punctae is somewhat confusing-one* can *only see this structure if the diffuse pool was effectively degraded, and all of the mutants basically need to do this in order for the authors to capture differences in β-cat accumulation within punctae. I think it would be helpful to clarify that all of the mutants* can *degrade β-catenin (as evidenced by depletion of the diffuse cytosolic pool of β-cat), but that there must be rate differences due to accumulation differences in the punctae.*

This is a good point. The photoconversion experiment would ultimately help to clarify this, and we now mention this in the Discussion as a useful future direction. We also clarify in the Discussion that, as noted, many of the mutants can stimulate ßcat degradation (at least when expressed together with Axin), and thus we must be looking at a rate difference. The relevant text in the Discussion now reads: “Thus while Axin can template βcat phosphorylation and can, at least in the presence of the truncated APC1 present in tumor cells, send it on to destruction, our data suggest APC promotes the rate at which ßcat is transferred out of the destruction complex and sent to the proteasome. Future use of photoactivatible ßcat constructs will further clarify this.” And “Based on our model APC mediated βcat transfer is only possible when R2 and B are maintained in the truncated APC (Figure 10). R2 and B are essential for function in the absence of endogenous APC function in vivo in *Drosophila* (41). In SW480 cells our data suggest they are essential for APC2 to further simulate the rate of ßcat destruction mediated by Axin transfection.”

*10) It would be helpful if*
Figure 1
*showed that domain B is also called the Catenin Interaction Domain (CID,*
[6]*), and clearly define this since CID was not adequately described in the Introduction. Also, since all reviews cite that Axin binds CK1 in addition to GSK3, it would be helpful to have this interaction mapped for completeness (if precisely known)?*

Thanks for these suggestions. We had included the CID designation in Figure 1 but somehow left it out of Figure 1, and we had not done full justice to the CID in the Introduction. We have fixed the Figure as suggested and added the following to the Introduction:

“In exploring APC's mechanism of action when Wnt signaling is inactive, we recently focused on two conserved binding sites in APC, 20R2 (R2) and motif B (B; this is also known as the catenin interaction domain = CID; [22]; [41]; [46]; [6]; Figure 1). R2 is related to the 20R βcat binding sites, but lacks a key interacting residue and cannot bind βcat (29). B is immediately adjacent to R2 – its function was initially unknown, though it was recently shown to bind α-catenin, and thus play a role in Wnt regulation (6).

Strikingly, although other 20 Rs are individually dispensable, both R2 and B are essential for the destruction complex to target βcat for destruction (22; 41; 46). Our data further suggested that R2/B negatively regulates APC/Axin interaction, a somewhat surprising role for an essential part of the destruction complex.”

We also have added to the diagram in Figure 1 the region defined by Reinhardt et al. (2007) as sufficient to bind CK1 (it overlaps with the PP2A binding region).

*11) In the fourth paragraph of the Introduction it sounds like the Arm domain of APC behaves as an intrinsically disordered domain, but it seems more appropriate to use this description for the 15 and 20R downstream (especially since many β-cat binding ligands are ID in the region of binding. Please clarify or specify the regions in*
Figure 1
*as “intrinsically disordered region, or IRD.”*

Thanks for this suggestion. We now have labeled Figure 1 to make it clear that the intrinsically disordered region is that downstream of the Arm repeats, containing the 15 and 20 AA repeats, the SAMPS, R2 and B. We also have made this clearer in the text as follows: “APC combines an N-terminal Armadillo repeat domain (Arm rpts) with a more C-terminal intrinsically disordered region; each contains binding sites for multiple partners (Figure 1).”

12) To study the dynamics of APC-Axin interactions, the authors transfect various fly and human constructs into the well-known SW480 colon cancer cell line, which expresses a wild type copy of APC and a truncation of APC containing the first 1338 AA. In their analyses of Axin-APC interactions, they do not take into consideration the effect of endogenous proteins in mediating the interaction. The role of endogenous proteins is only briefly mentioned in the subsection “APC2 stabilizes the Axin complex and promotes DC throughput of βcat”.

This is an important issue, and, as the reviewers points out, it needs to be made clear as these endogenous proteins may well be playing important roles (we assume there was a typo in the Review comments and that the Reviewer was referring to either the wildtype copy of APC2 or the wild-type copy of Axin expressed in these cells – they only express a truncated APC1).

We have addressed this in two ways. First, our data showed that expression of either “APC2 alone” or APC2 plus Axin similarly stimulated ßcat exit out of the puncta to destruction. In our initial manuscript, we had suggested this was “presumably due to the presence of endogenous human Axin”. We now have tested whether fly APC2 can and does interact with human Axin. We found that Fly GFP-APC2 colocalizes in puncta with exogenous hAxin1-RFP (present in new Figure 3), and more importantly, that endogenous hAxin1 coIPs with Flag-APC2 expressed in Sw480 cells (new Figure 3). Thus this is at least a reasonable possibility.

Second, we significantly expanded our description of the status and potential roles of endogenous proteins in SW480 cells. We now begin the Results section with a more complete description of this, as follows: “Here we sought to define the mechanism by which these motifs and APC itself act, using as a model SW480 colon cancer cells. These cells have high βcat levels, as they lack wildtype human APC1 (hAPC1) and instead express a truncated APC1 protein ending before the mutation cluster region (MCR; Figure 1). SW480 cells also express human APC2 (31), but this is not sufficient to help mediate βcat destruction.” We note the potential importance of truncated APC1 and APC2 in the residual function of Axin over-expression in these cells when we initiate examination of Axin vs Axin + APC2 function, as follows: “Axin cannot target βcat for destruction in APC's complete absence, even when Axin is overexpressed (34). However, in SW480 cells, which express both truncated hAPC1 and endogenous hAPC2 (31), Axin overexpression can increase βcat destruction (35).” Finally, we now end the relevant section of the Discussion by pointing toward the importance of further resolving these issues in the future, as follows: “The development of CRISPR knockout technology will allow future examination of the importance of truncated APC1, as well as the endogenous APC2 and Axin expressed in colon cancer cells in destruction complex assembly and function.”

*13) In*
Figure 8
*the authors identified putative GSK3 phosphorylation sites. It is unclear how these meet the consensus for GSK3 other than being Ser/Thr residues. Usually there is a priming site 4-5 amino acids C terminal to the putative target site. The authors need to define what criteria they used to identify putative sites. I have a number of concerns about this experiment:*

The reviewers made a number of good points, each of which we have addressed.

The kinase assay is lacking positive controls.

This was a good point – we had done a positive control with the GSK3 substrate peptide provided by the manufacturer and this is now included in Figure 8.

What is the band in the GST/GSK3 lane?

When we incubated GSK3 with GST (not an expected substrate) we saw a labeled band consistent with autophosphorylation. We now note this in the Figure legend. This was not seen when GSK3 was incubated with a good substrate (e.g., hAPC1R2/B or even fly APC2R2/B). We observed this in both of our experiments (we now show a different experiment than that previously used to deal with the issue pointed out below).

Why are there multiple bands in the R2/B/GSK3 lane?

In both independent experiments there was a smaller Coomassie-stained band in the hAPC1R2/B purification that we presume is a breakdown product – this protein is also phosphorylated.

What do the asterisks mean in the figure?

These are the Coomassie bands that align with phosphorylated proteins – we now explain this in the Figure Legend.

Why are there GST-only bands in the coomassie lanes 3-6?

These were apparently background bands from the purification. We have now replaced the previous blots with our second experiment in which these bands are no longer present.

What was the rationale for mutating the two selected Ser residues? They don't seem to match the consensus any better than the others. The logic given states that they are the most N terminal, when in fact two other Ser residues are more N terminal, and there are no potential CK1 priming sites.

The reviewers correctly point out that this logic is faulty, though it does reflect the spirit of the thought process we went through. The choice of these residues has historical roots that are somewhat complex. We initially aligned APC sequences from a number of different species, including several different vertebrates and arthropods. The reviewer correctly points out that there are two more serines in “motif B” N-terminal to those we mutated. However, in fly APC1, fly APC2, and the single APC family member in the brine shrimp *Daphnia*, there is a cysteine residue deleted relative to the vertebrate APC1 or APC2 sequences, and thus the spacing of these more N-terminal serines was not exactly the same. As a result, one of these N-terminal serines doesn't share the correct spacing in the arthropod family members to be a match to the usual GSK3 consensus (S-X-X-X-S) (we now make this clear in the Figure). We thus initially confined our analysis to the region of motif B C-terminal to this deleted residue. We began with this more confined definition of motif B, and mutated two, four or six of the conserved serine residues to alanine. Each of these essentially eliminated function of APC2 in downregulating ßcat levels. We thus continued our analysis with the least altered of these, which altered the two residues on which we focused in our work. We now explain this more clearly in the text and remove the previous “faulty logic”, as follows:

“R2 and B are essential for APC2's function in βcat degradation. Our hypothesis suggests phosphorylation of R2/B promotes release of APC2's Arm rpts from the Axin complex, and that this would be essential for the catalytic cycle of the destruction complex – thus mutating these putative GSK3 phosphorylation sites in motif B would reduce APC2 function in helping mediate ßcat destruction. To begin to test this, we replaced 2 (APC2AA; Figure 8, red arrows), 4 or 6 (Figure 8, magenta arrowheads) conserved serines in B that match the GSK3 consensus with alanine, to prevent phosphorylation.

All reduced function in downregulating ßcat levels (see below and data not shown). We thus focused on the least altered of these, the mutant that replaced the more N-terminal two serine residues with alanine (APC2AA; Figure 8, red arrows), thus preventing phosphorylation. We also created a mutant that replaced these same two residues with aspartic acid, creating a phosphomimetic APC2 (APC2DD).”

We also added additional annotations to Figure 8, making all of this clearer. We'd also note that we actually missed a match to the CK1 consensus near the C-terminus of motif B – we now have added this to Figures 7 and 8.

As the authors wisely state in subsection “Mutating putative phosphorylation sites in B disrupts APC2's function in regulating βcat destruction in SW480 cells”, it is very likely that GSK3 inhibition affects many aspects of DC dynamics, so all findings should be interpreted cautiously. The Drosophila work with the phospho-resistant and phospho-mimetic transgenes is very elegant.

This is also an important point. As we describe in detail above in response to reviewer point 3 above about the link between phosphorylation and dynamics, in the initial version of the manuscript we went too far in trying to firm up what was an admittedly speculative link. We have tried to tone this down as recommended here. We now emphasize that two different mutations of conserved serines (the previously described mutations in motif B and the new mutation we describe here in R2) all strongly reduce APC2 activity in SW480 cells, and in the case of the double point mutant in motif B, also do so in *Drosophila*. However, as the Reviewer correctly points out, none of our point mutants match in severity the effect on dynamics or function of either GSK3 inhibition or deletion of either R2 or B. We thus have tried to make this clearer in both the Results and Discussion.

For example, we now conclude this section of the Results as follows: “Together these data are consistent with the idea that phosphorylation of conserved serine residues in R2/B regulates APC's function in the destruction complex, but since the effect on dynamics was substantially less dramatic than that of deleting R2 or B, it suggests other residues in R2 and B may also contribute to regulating APC2 dynamics. Further, it is clear that GSK3 has other effects on the complex, complicating interpretation of its inhibition. It will be important to examine this in the future.”

And in the Discussion, state: “Our data are consistent with a model in which GSK3 phosphorylation of R2 and B could be one major regulatory step in APC2's dynamic cycle in the destruction complex, triggering release of the Arm rpts from the Axin complex, and thus allowing APC2 to promote βcat release for destruction (Figure 10). However, our data also show that mutating two residues in motif B to prevent their phosphorylation had only a subtle effect on APC2 dynamics. This may suggest additive roles for multiple phosphorylated residues, or may suggest the connection between phosphorylation, dynamics and function is more complex. Further, GSK3 can phosphorylate most of the other proteins in the destruction complex, and thus it clearly plays multiple roles in its function.”

14) In their model, as described in the Discussion, the authors suggest that GSK3 and CK1 phosphorylate R2 and B motifs. They cannot make this conclusion, since B has no CK1 motifs and no kinase assays with CK1 were performed.

The point is a good one. We did try in vitro phosphorylation with purified CK1 and there was phosphorylation of human APC1 R2/B, but the phosphorylation of fly APC2 R2/B was very weak at best. We thus do not have any evidence CK1 is involved in R2/B regulation. We thus have altered the text as follows (and made parallel changes in Figure 10):

“GSK3, which phosphorylates βcat, Axin and other sites on APC, also phosphorylates APC2's R2 and B (step 3; CK1 may also be involved, as it is in the other phosphorylation events in the complex).”

As noted above, we missed a match to the CK1 consensus near the C-terminus of motif B – we now have added this to Figures 7 and 8.

*They also state that Axin assembles in a head to tail fashion* via *its DIX domain. I would argue this is a tail to tail interaction.*

We apologize for our confusing description. The head-to-tail description of Axin polymerization comes from the original publication – in this case the authors refer to the isolated DIX domain, in which beta-sheet 2 of one DIX domain interacts with beta-sheet 4 of the next monomer, thus making it “head-to-tail” with respect to the two ends of the monomer. However, as the DIX domain is at the Cterminus (the tail) of the whole protein, we see how this caused confusion. We now have clarified this as follows (this also now makes the diagram in Figure 10 clearer):

“One mechanism involved was already known: the DIX domain of Axin polymerizes in a head-to-tail fashion, in which beta-sheet 2 of one DIX domain interacts with beta-sheet 4 of the next monomer, thus forming filaments, and this polymerization is essential for destruction complex function (47; 13).”